# Single-molecule imaging reveals replication fork coupled formation of G-quadruplex structures hinders local replication stress signaling

Wei Ting C. Lee[1], Yandong Yin [1], Michael J. Morten [1], Peter Tonzi[1], Pam Pam Gwo[1], Diana C. Odermatt[2], Mauro Modesti [3], Sharon B. Cantor[4], Kerstin Gari [2,5], Tony T. Huang[1] & Eli Rothenberg [1✉]

Guanine-rich DNA sequences occur throughout the human genome and can transiently form G-quadruplex (G4) structures that may obstruct DNA replication, leading to genomic instability. Here, we apply multi-color single-molecule localization microscopy (SMLM) coupled with robust data-mining algorithms to quantitatively visualize replication fork (RF)-coupled formation and spatial-association of endogenous G4s. Using this data, we investigate the effects of G4s on replisome dynamics and organization. We show that a small fraction of active replication forks spontaneously form G4s at newly unwound DNA immediately behind the MCM helicase and before nascent DNA synthesis. These G4s locally perturb replisome dynamics and organization by reducing DNA synthesis and limiting the binding of the single-strand DNA-binding protein RPA. We find that the resolution of RF-coupled G4s is mediated by an interplay between RPA and the FANCJ helicase. FANCJ deficiency leads to G4 accumulation, DNA damage at G4-associated replication forks, and silencing of the RPA-mediated replication stress response. Our study provides first-hand evidence of the intrinsic, RF-coupled formation of G4 structures, offering unique mechanistic insights into the interference and regulation of stable G4s at replication forks and their effect on RPA-associated fork signaling and genomic instability.

[1] Department of Biochemistry and Molecular Pharmacology, New York University School of Medicine, New York, NY, USA. [2] Institute of Molecular Cancer Research, University of Zurich, Zurich, Switzerland. [3] Cancer Research Center of Marseille, CNRS UMR7258, Inserm U1068, Institut Paoli-Calmettes, Aix-Marseille Université, Marseille, France. [4] Department of Molecular, Cell and Cancer Biology, University of Massachusetts Medical School, Worcester, MA, USA. [5] Present address: Institute of Chemistry and Biotechnology, Zurich University of Applied Sciences, 8820, Wädenswil, Switzerland. ✉email: Eli. Rothenberg@nyumc.org

The human genome contains 370,000–700,000 repetitive guanine-rich sequences that have the potential to spontaneously fold into stable G-quadruplex (G4) structures under physiological conditions[1,2]. These non-canonical DNA secondary structures are formed by the stacking of several G-quartets, which are square planar structures formed by four guanine bases stabilized through Hoogsteen hydrogen-bonding. Several G-quartets stack on top of one another to form a four-stranded helical G4 structure that is further stabilized by monovalent cations such as $K^+$ and $Na^+$ [3–6]. G4 structures have been proposed to serve as regulatory elements for DNA replication, transcription, and telomere regulation[7–10], yet their existence has also been linked to mutagenesis[11–14]. In particular, many of the identified G4-motifs were mapped within oncogenes, as well as point mutations, translocation breakpoints, indels, and copy number variations that are frequently found in cancers[15,16], leading to the hypothesis that deregulated G4 formation may act as physical obstacles for DNA metabolisms such as replication. However, our knowledge of and mechanistic insights into the occurrence and regulation of G4s, as well as their effects on the replication machinery, remain nominal.

The conceivable biological significance of G4s has made them an important area of research and a potentially tractable therapeutic target[17,18]. Nevertheless, understanding the effects of these structures in vivo has been challenging because their formation can be highly dynamic and transient, thus requiring the development of targeted reporter assays. Recent efforts used to study the effects of G4s on replication include the use of replicating plasmids with G4 sequences, monitoring genomic instability at specific genomic loci that contain putative G4-motifs, the use of reporter assays, and genome wide deep-sequencing[13,14,19,20]. These approaches identified important in vivo regulatory roles of both G4 motifs as well as various G4-interacting proteins and helicases, demonstrating that persistent and/or deregulated G4s pose an impasse to replication fork progression, causing genetic and epigenetic instability[15,21]. Recent development of antibodies and ligands to probe for G4s have provided compelling visual evidence of their formation in vivo[22,23]. Importantly, the amount of G4s was found to be elevated in various human cancer cell lines particularly in S phase[22,23]. These observations suggest that transient unpackaging of chromatin and exposure of ssDNA during replication could be conducive to G4 formation, hence posing a moment of heightened replisome vulnerability[24–27]. Despite these studies, we still lack direct evidence as to the formation of replication fork (RF)-coupled G4, which are hypothesized to spontaneously occur during replication of the numerous endogenous G4 motifs within our genome. Consequently, we have a poor understanding regarding how these structures directly affect replication fork activity, morphology, and signaling in vivo. This is, in part, due to previous technical limitations in the ability to visualize transient RF-coupled G4s in vivo.

To address this knowledge gap, we utilized multi-color single-molecule localization microscopy (SMLM) for direct, nanoscopic visualization of replication factors, nascent DNA, and G4s in cells. To obtain an unbiased quantitative classification of the spatial patterns of different molecular complexes resolved within each cell, we developed a robust SMLM image data mining algorithm[28,29]. We utilized this method to quantify the association of G4s with individual replisomes, and observed the formation of G4s at a subset of active replication forks that is further enhanced by the induction of helicase-polymerase uncoupling. Formation of G4s within replisomes imposes distinct replication dynamics, impeding DNA polymerase progression and hindering the recruitment of Replication Protein A (RPA), a ssDNA-binding protein essential for replication stress signaling.

Suppression and resolution of RF-coupled G4 formation is mediated via the collaboration between replication-associated helicase FANCJ and RPA; without FANCJ helicase activity, local accumulation of stable G4s at forks dampens RPA-mediated replication stress signaling, culminating in DNA damage at corresponding replication forks. Our single-molecule fluorescence resonance energy transfer (smFRET) assays further identified the interplay between FANCJ and RPA, in which FANCJ helicase activity facilitates the loading of RPA onto thermodynamically stable G4 structures that are otherwise refractory to RPA binding. Collectively, our results provide a novel mechanistic understanding of the formation, consequence and regulation of stable G4s that form during DNA synthesis in the context of replisome organization and dynamics, as well as replication fork protection and signaling.

## Results

**Quantitative single-molecule localization of RF-coupled G4s in cells.** To visualize individual replisome complexes and their association with DNA G4 structures in cells, we utilized recently established multi-color SMLM imaging protocols[30,31] wherein samples were labeled with photoswitchable dyes. We pulse-labeled nascent DNA with the thymidine analog 5-ethynyl-2′-deoxyuridine (EdU), and co-stained with antibodies against the replicative helicase MCM, DNA polymerases processivity factor PCNA (Fig. 1a) or DNA G4 (Fig. 1d) (see antibody validation in Supplementary Note 1). We then selected EdU-positive, S phase nuclei for imaging. In contrast to blurry images typically obtained from diffraction-limited microscopy, the enhanced single-molecule detection sensitivity of SMLM provided nanoscale localization and resolution of the labeled molecules, with images representing the molecular coordinates of all localized molecules (Fig. 1a, d).

To estimate the frequency of replication sites that encounter G4 structures, we first measured the fraction of PCNA foci that are non-randomly colocalize with G4s by utilizing quantitative SMLM clustering (density-based spatial clustering of applications with noise (DBSCAN)) and colocalization (nearest neighbor distance (NND)) approaches[32]. This revealed that a small, yet statistically significant, subset (~2.24%) of the observed replication sites form G4 structures (Supplementary Fig. 1j, Supplementary Note 2). We next sought to resolve the specific molecular arrangement of G4s at replication forks, and to examine how their formation affects the organization and dynamics of individual replisomes. We noted that although the improved resolution (10–20 nm) of SMLM allows for localization of individual replisome components[33], the crowding of numerous replisomes within a nucleus complicates the identification and quantification of the molecular arrangements corresponding to individual replisomes or G4-associated replisomes. To address this issue, we applied an unbiased automated pattern recognition approach employing the Triple-Correlation (TC) function as previously described[28,29]. This is a robust data-mining algorithm that identifies triplet molecular patterns by calculating the spatial-correlation statistics among the coordinates of all localized molecules from each of the three channels within a single nucleus. Figure 1b, e shows the resulting TC-resolved molecular configurations (termed "TC Triplet") from the single nucleus in Fig. 1a, d, respectively (see detailed explanation on TC analysis, calibration, and modeling in Supplementary Note 3). These patterns represent the most probable and statistically significant configurations derived from the average of all triplet patterns identified within a nucleus. Accordingly, the TC triplet obtained from the SMLM image of EdU, MCM, and PCNA shows a unique consecutive EdU-PCNA-MCM arrangement, comparable to the

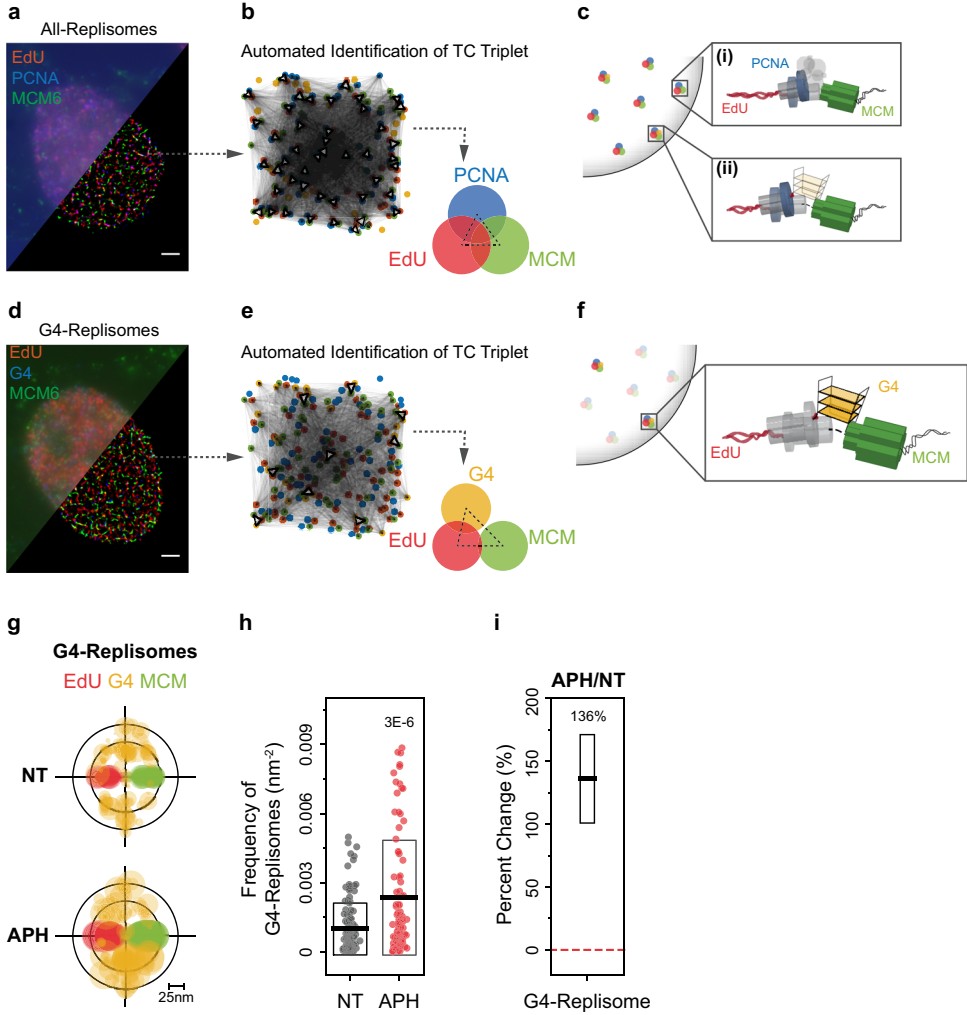

**Fig. 1 Direct observation and quantification of DNA G4 structures and their association with replisomes. a, d** Representative epifluorescence (upper left) and SMLM (lower right) images of a single S-phase U2OS nucleus labeled for **a** nascent DNA (using EdU, red), PCNA (blue), and MCM (green); and **d** nascent DNA (using EdU, red), G4 (blue), and MCM (green). Scale bar, 2 μm. **b, e** Schematic illustrations of how TC analysis recognizes triplet patterns from a nucleus. Each red, blue (yellow for G4), and green molecules can form triangles (connected by pale gray lines). If a specific pattern is repeatedly found (dark, bold triangles), its population is distinct from stochastic triplets and therefore is identified as a TC triplet, as shown on right. **b** shows the TC triplet derived from **a**, while **e** shows the TC triplet derived from **d**. **c, f** Schematic illustrations of the molecular organizations of "All-Replisome" (without (i) or with (ii) G4 association) resolved by TC analysis of EdU, PCNA, and MCM, as represented in **a** and **b** (**c**), and "G4-Replisome" resolved by TC analysis of EdU, G4, MCM, as represented in **d** and **e** (**f**). **g** Overlaid TC triplets of EdU, G4, and MCM from multiple non-treated (NT) or 1 h, 200 nM APH-treated cells statistically describe the molecular organization of these three species. Circle size of each TC triplet represent the frequency of G4-Replisomes from a single nucleus. The TC triplets are aligned onto the same EdU-MCM plane to define the positions of G4s relative to the replisome complex. **h** Frequency of G4-Replisomes in NT or APH-treated cells. Individual data points represent result from single cell. Black horizontal line and box height indicate mean ± SD. Values on graph indicate p-values of unpaired two-sample t-tests between NT and APH-treated cells. **i** Percent change in frequency of G4-Replisomes in APH-treated compared to NT cells. Values on the graph and black horizontal line represent the percent change, box height indicates the propagated s.e.m. For all experiments, number of cells analyzed and TC triplets identified are listed in Supplementary Table 1.

characteristic single-replisome assembly[28,29] with respect to their morphologies and scale[33,34]. We termed this configuration as "All-Replisomes" since it measures the entire population of replisomes within a nucleus (Fig. 1c).

Importantly, the obtained TC triplet patterns for the SMLM images of EdU, MCM, and G4 (Fig. 1e, g) revealed a significant association between G4 and replisomes. To test whether RF-coupled formation of G4s arises from ssDNA exposure within replisomes[15], we induced increased ssDNA stretches at replication forks by briefly (1 h) treating S-phase cells with a low concentration of the DNA polymerase inhibitor aphidicolin (APH) (Supplementary Fig. 1a). This mild treatment condition does not result in complete replication fork stalling and has

negligible effects on cell cycle checkpoint responses[35]. We hypothesized that increased ssDNA exposure within replisomes due to reduced polymerase progression and continued MCM unwinding activity would in turn enhance RF-coupled G4 formation. Indeed, APH treatment led to a substantial increase in the frequency of G4s that are associated with MCM and EdU compared to NT cells, as quantified by TC analysis (Fig. 1g–i), and independently by the DBSCAN/NND approach (Supplementary Fig. 1j), with a corresponding increase in nuclear G4 signals, calculated by autocorrelation (AC) analysis[36] (Supplementary Fig. 1b). Together, our observations provide strong evidence that DNA G4s can spontaneously form during replication fork progression as double-stranded DNA (dsDNA) is

unwound into ssDNA (Fig. 1f). We emphasize that this association specifically represents RF-coupled G4s, or the sub-population of replisomes that form stable G4s during unwinding, which we term "G4-Replisomes".

**RF-coupled G4 formation impedes DNA synthesis and counteracts RPA binding**. RPA plays major roles in safeguarding replication forks by binding and protecting ssDNA that is exposed during replication[37]. We therefore asked whether RPA is also associated with the G4-Replisomes we observe in cells. To this end we performed SMLM imaging along with AC and TC analyses to measure the extent of RPA within All-Replisomes (RPA/MCM/PCNA) compared to the level of RPA in G4-replisomes (RPA/MCM/G4) upon APH exposure (Supplementary Fig. 1f). In agreement with our previous report[29], APH treatment resulted in an increase in nuclear RPA signals (Supplementary Fig. 1g, calculated by AC), as well as a specific increase in RPA within All-Replisomes (Fig. 2a–c, Supplementary Fig. 1h, "All-Replisomes", calculated by TC), corresponding to the increase in ssDNA exposure upon helicase-polymerase uncoupling induced by APH. Surprisingly, analysis of local RPA level at

G4-Replisomes showed no substantial increase even following APH treatment (Fig. 2a–c, Supplementary Fig. 1i, "G4-Replisomes"). We infer that RF-coupled formation of G4s locally hinders RPA binding at the fork (Fig. 2d).

We also examined whether G4 formation affects replication fork progression by quantifying the amount of EdU signals at individual replisomes during normal replication and upon APH treatment (Supplementary Fig. 1a). As expected, APH treatment led to a global reduction in EdU incorporation at all forks (Supplementary Fig. 1c–e). Notably, EdU incorporation at G4–Replisomes was consistently lower than at All-Replisomes, regardless of APH treatment (Fig. 2f, g). These observations establish that RF-coupled G4 formation locally disturb their replication behavior, resulting in reduced RPA binding and a diminished rate of DNA synthesis (Fig. 2d, h).

**Regulation of RF-coupled G4 formation**. We next sought to elucidate the mechanism by which G4s within the replisome are regulated. A well-supported candidate for this role is the FANCJ DNA helicase encoded by *BRIP1* (BRCA1 interacting protein C-terminal helicase 1) gene[38], because FANCJ has specific in vitro

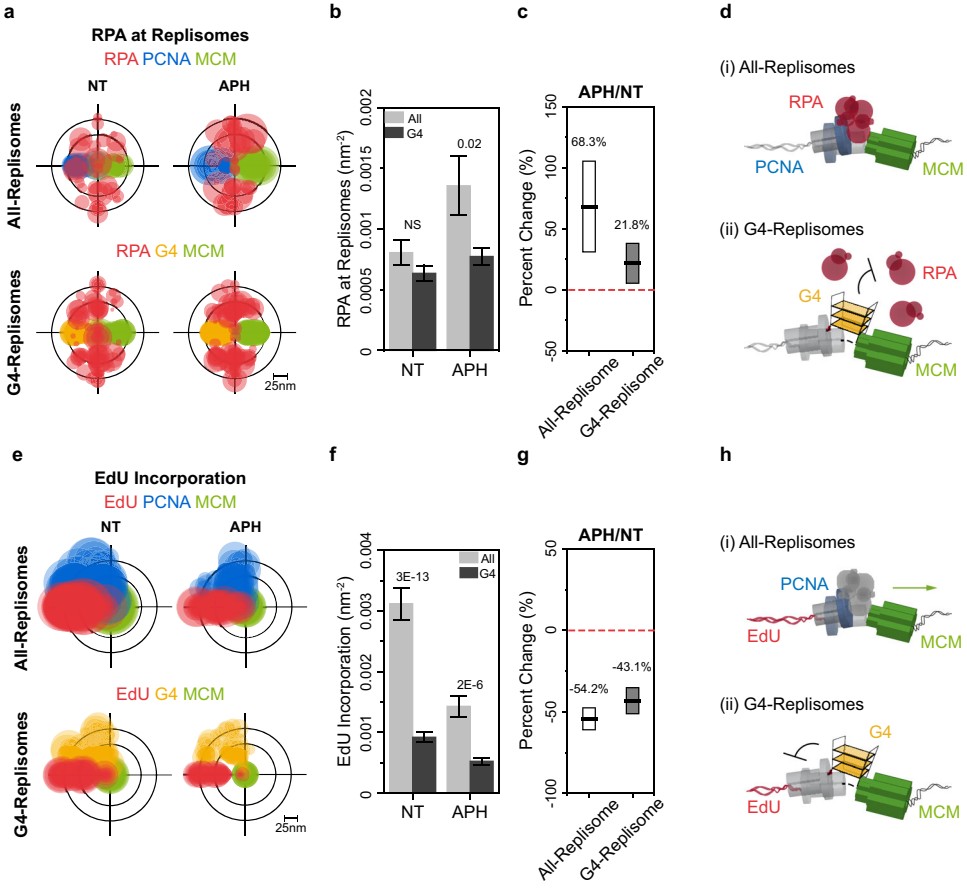

**Fig. 2 Effects of G4 formation on replisome structure and progression. a**, **e** Overlaid TC triplets of RPA, PCNA, MCM (**a**) or EdU, PCNA, MCM (**e**) (top, All Replisomes), and RPA, G4, MCM (**a**) or EdU, G4, MCM (**e**) (bottom, G4-Replisomes) from multiple NT or APH-treated cells. Circle size of each TC triplet represents the local density of RPA (**a**) or EdU (**e**) from a given nucleus. For **a**, the TC triplets are aligned onto the same EdU-MCM (top) or G4-MCM (bottom) plane to define the positions of RPA relative to the replisome complex. For **e**, the TC triplets are aligned using MCM as the center to better visualize the relative magnitude of EdU. **b**, **f** Comparison of the local densities of RPA (**b**) or EdU (**f**) within All-Replisomes (light gray) or G4-Replisomes (dark gray) in NT and APH-treated cells. Error bars indicate mean ± s.e.m. Values on graph indicate *p*-values of unpaired two-sample *t*-tests between All-Replisomes and G4-Replisomes. Corresponding data plots showing the data distributions are presented in Supplementary Fig. 1d, e, h, i. **c**, **g** Percent change in the densities of RPA (**c**) or EdU (**g**) at All-Replisomes or G4-Replisomes in APH-treated compared to NT cells. Values on the graph and black horizontal line represent the respective percent changes, box height indicates the propagated s.e.m. **d**, **h** Schematic illustrations showing that the spontaneously folded G4 structure at an active replication fork blocks RPA recruitment onto ssDNA during regular replication (**d**) and locally hinders DNA synthesis (**h**). For all experiments, number of cells analyzed and TC triplets identified are listed in Supplementary Table 1.

G4 unwinding activity and functions in replication fork protection[14,19,20,23,39,40]. To determine the localized response to RF-coupled G4 formation, and the contribution of FANCJ to this process, we briefly treated U2OS cells (siCTRL or siFANCJ) with the G4-stabilizing ligand pyrodistatin (PDS)[22,41–43], then selected EdU- and/or PCNA-positive S-phase cells for imaging and analysis (Supplementary Fig. 2a). Depletion of FANCJ in U2OS cells resulted in an enrichment of nuclear G4s (Supplementary Fig. 2b), which was further elevated upon PDS exposure. In contrast, the brief PDS treatment did not yield a noticeable change in nuclear G4 signal in siCTRL cells (Supplementary Fig. 2b). To determine whether the observed increase in G4 structures stems from the stabilization of RF-coupled G4s, we quantified (via TC analysis) the relative G4-Replisome frequencies in siCTRL and siFANCJ cells upon PDS stabilization. This revealed that PDS treatment in siFANCJ cells resulted in a substantial increase of G4-Replisomes, whereas PDS treatment in siCTRL cells only caused a modest increase (Fig. 3a–c). These results are consistent with data obtained in a CRISPR-mediated FANCJ knockout (FANCJ-KO) HeLa cell-line with or without wild-type FANCJ (FANCJ-WT) or a helicase-dead FANCJ$^{K52R}$ (FANCJ-HD) complementation[44]. The observed trends were further confirmed using our DBSCAN/NND approach (Supplementary Fig. 2c). Combined, these data demonstrate that the resolution of stable G4s that form at replication forks requires the helicase activity of FANCJ.

Since RF-coupled G4 formation can induce local replication fork blockade (Fig. 2), we hypothesized that the increased G4-Replisome frequency in FANCJ-deficient cells would also lead to a reduction in replication progression. We therefore measured the abundance of EdU signal at replisomes as an indication of replication fork progression. Indeed, the increased G4-Replisome frequency in siFANCJ cells upon PDS treatment (Fig. 3a–c) was accompanied by a reduction in EdU at both nuclear level (Supplementary Fig. 2d, e) and individual replisome level (EdU/MCM/PCNA) (Fig. 3d–f), whereas the level of EdU incorporation in siCTRL cells remain unchanged upon PDS treatment (Fig. 3d–f; Supplementary Fig. 2d, e). Importantly, EdU incorporation at G4-Replisomes (EdU/MCM/G4) was consistently lower than that of All-Replisomes regardless of treatment conditions (Fig. 3g–i). Taken together, our results demonstrate that RF-coupled G4 formation presents an obstacle to replication fork progression and such events are further exacerbated in the case of FANCJ deficiency.

Recruitment and phosphorylation of RPA at stalled replication forks is a hallmark of the ATR-mediated replication stress response[45,46], whereas the transient, RF-coupled G4 formation limits RPA binding despite helicase-polymerase uncoupling (Fig. 2). We therefore investigated RPA's recruitment and signaling in response to RF-coupled G4 formation, and how these are affected by stable G4 accumulation at replisomes due to FANCJ depletion (Fig. 3a–c). Analysis of the relative levels of RPA at All-Replisomes (RPA/MCM/PCNA) and at G4-Replisomes (RPA/MCM/G4) in siCTRL cells revealed a substantial enrichment in RPA binding at All-Replisomes following 4 h of PDS treatment, while siFANCJ cells showed no significant change (Fig. 4a–c; Supplementary Fig. 3a, c, d). This data indicates that a direct response to RF-coupled G4 formation involves an increase in FANCJ-mediated RPA loading. In contrast, RPA levels at G4-Replisomes remain consistent irrespective of PDS treatment (Fig. 4d–f; Supplementary Fig. 3b). The limited loading of RPA at G4-Replisomes is in line with the diminished EdU incorporation upon PDS stabilization (Fig. 3) and our observations in APH-treated cells (Fig. 2). To further substantiate our findings, we examined the effects of G4 accumulation on RPA recruitment in the FANCJ-KO HeLa cells

with or without FANCJ-WT or FANCJ-HD complementation. In agreement with our observations in U2OS cells, we found an increase in RPA loading at All-Replisomes following PDS treatment in FANCJ-WT cells, which was otherwise restrained in FANCJ-KO and in FANCJ-HD cells (Supplementary Fig. 3e, f). Combined, these results indicate that G4 accumulation within replisomes can be suppressed by rapid loading of RPA onto unwound G4 structures which requires FANCJ helicase activity.

**Binding of RPA onto stable short-loop G4s is facilitated by FANCJ.** Our observations whereby FANCJ depletion corresponds to limited recruitment of RPA to forks along with an increased frequency of RF-coupled G4 formation suggest that the binding of RPA onto G4s at forks requires FANCJ activity. Previous in vitro studies have shown that FANCJ interacts with RPA to stimulate DNA unwinding activity[20,47], whereas other reports have demonstrated that RPA can directly bind and destabilize pre-formed G4 structures in vitro in a sequence and structure-stability-dependent manner[48–50]. To gain further mechanistic insight into the roles of RPA and FANCJ during DNA G4 resolution, we used a smFRET assay to measure the real-time unfolding of G4 structures in the presence of either RPA or FANCJ, or both proteins, in vitro. In this assay, a donor–acceptor FRET pair is positioned at either side of an intra-stranded G4-forming sequence so that a folded G4 brings the fluorophores into close proximity and results in a high FRET efficiency ($E_{FRET} =$ 0.7), while destabilization of the G4 structure increases the distance between the FRET pair and lowers the observed FRET efficiency ($E_{FRET} = 0.25$) (Fig. 5a, b). A previous in vitro study has shown that the enhanced thermostability of short-loop G4s can limit RPA binding and unfolding[50], while a recent computational analysis revealed an over-representation of single-nucleotide-loop G4 motifs in the human genome[51]. To probe the contribution of intrinsic G4 stability, we designed two G4 constructs having the same G-tetrad layers and ssDNA overhangs but connected with either 1- or 3-nucleotide long loops (G4 L1 and G4 L3, respectively, Supplementary Table 3).

We first examined the ability of RPA alone to bind and unfold these structures by monitoring the FRET efficiencies of pre-folded G4 constructs in the presence of increasing concentrations (0–20 nM) of RPA. These measurements revealed transient G4 unfolding events observed in smFRET trajectories (Fig. 5b) and showed an emergence of unfolded G4 (low FRET peaks) along with a decrease in the fraction of folded G4 (high FRET peaks) with increasing RPA concentrations (Fig. 5c, d, Supplementary Fig. 4). In agreement with a previous study[50], RPA-mediated destabilization was observed for G4 constructs with longer loops (G4 L3) while the 1-nucleotide loop structure (G4 L1) did not display any persistent unfolding by RPA.

Next, we added FANCJ (100 pM) and ATP (1 mM) together with RPA, resulting in a dramatic destabilization of the folded G4 structures, indicating that FANCJ amplifies RPA-mediated G4 destabilization (Fig. 5b–d, Supplementary Fig. 4). We emphasize that the low concentration of FANCJ used in our measurements was chosen in order to specifically probe the contribution of FANCJ to RPA-mediated G4 unfolding, and to limit effects caused by FANCJ over-saturation. Accordingly, at this concentration of FANCJ (100 pM) we did not observe any persistent G4 unfolding, nor significant dsDNA unwinding, when RPA was omitted from the reaction (Fig. 5b), in agreement with previous studies[52]. In contrast, we found that this low concentration of FANCJ is sufficient for facilitating RPA's persistent destabilization of G4 L1. This unfolding requires the helicase activity of FANCJ, as no significant unfolding was observed in the presence of 10 nM RPA for FANCJ-HD + ATP, or with wild-type FANCJ

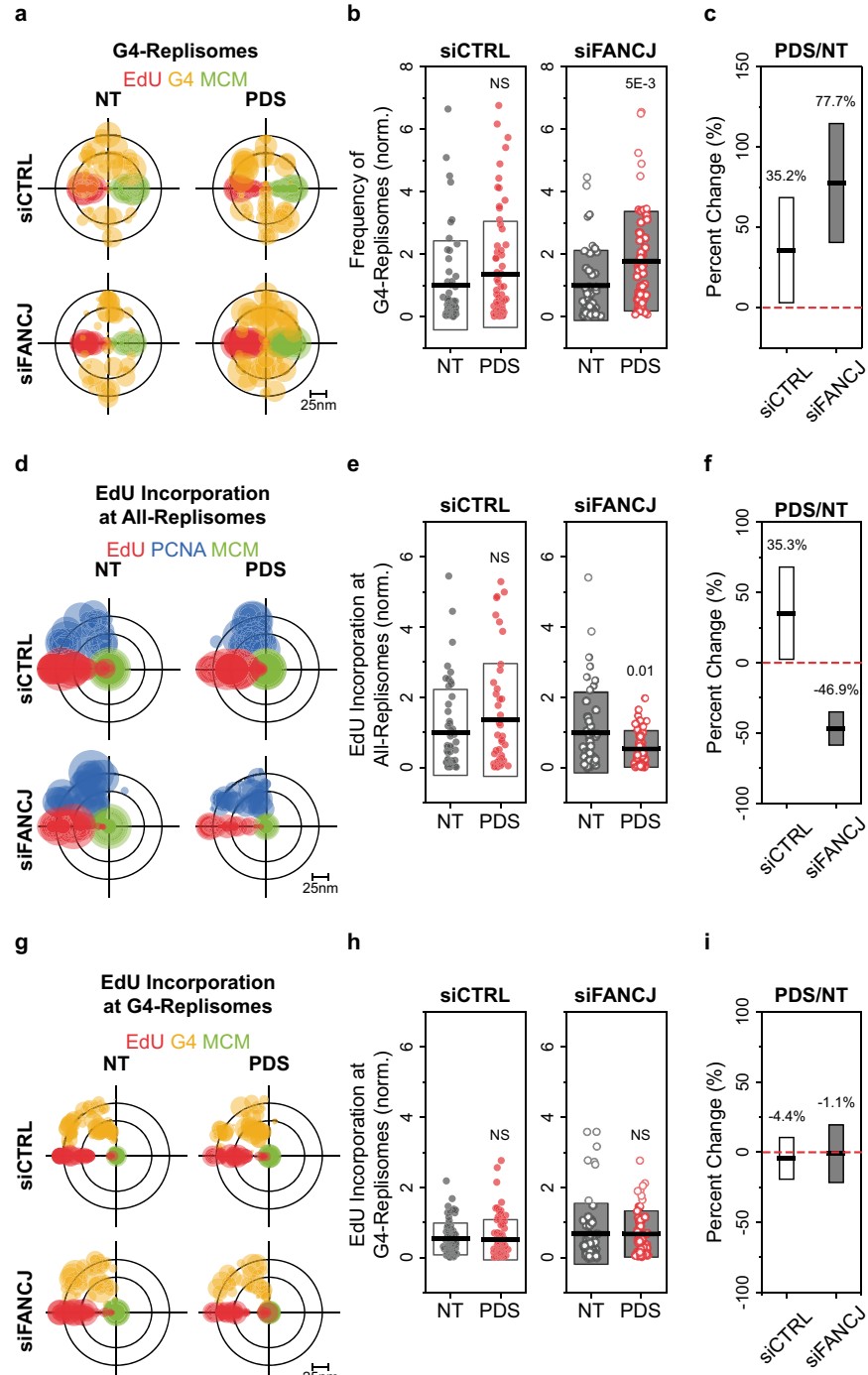

**Fig. 3 FANCJ suppresses accumulation of G4 structures within replisomes. a, d, g** Overlaid TC triplets of EdU, G4, MCM (**a**), EdU, PCNA, MCM (**d**), or EdU, G4, MCM (**g**), from multiple NT or 1 h, 20 μM PDS-treated S-phase U2OS cells transfected with control (siCTRL) or FANCJ (siFANCJ) siRNA. Circle sizes of each TC triplet represents the local density of G4-Replisomes (**a**), EdU at All-Replisomes (**d**), or EdU at G4-Replisomes (**g**) from a given nucleus. For **a**, the TC triplets are aligned onto the same EdU-MCM plane to define the positions of G4 relative to the replisome complex; for **d, g**, the TC triples are aligned using MCM as the center to better visualize the relative magnitude of EdU. **b, e, h** Frequencies of G4-Replisomes (**b**), EdU at All-Replisomes (**e**), or EdU at G4-Replisomes (**h**) in NT or PDS-treated siCTRL or siFANCJ cells. Individual data points represent result from a single nucleus. Black horizontal line and box height indicate mean ± SD. Values on graph indicate *p*-values of unpaired two-sample *t*-tests between NT and PDS-treated cells. **c, f, i** Percent change in the densities of G4-Replisomes (**c**), EdU at All-Replisomes (**f**), or EdU at G4-Replisomes (**i**) in siCTRL or siFANCJ PDS-treated compared to NT cells. Values on the graph and black horizontal line represent the respective percent changes, box height indicates the propagated s.e.m. For all experiments, number of cells analyzed and TC triplets identified are listed in Supplementary Table 1.

when ATP was omitted (Fig. 5e). Moreover, in the presence of RPA + ATP, FANCJ was able to disrupt both G4 L1 and G4 L3 to similar extents (Fig. 5d; Supplementary Fig. 4b), supporting the essential role of FANCJ in resolving otherwise stable DNA

secondary structures. For the more stable G4 L1, the observed G4 unfolding rate increased in the RPA + FANCJ + ATP condition compared to RPA alone, along with a decrease in the G4 folding rate (Fig. 5d). This trend to some extent was mirrored in the case

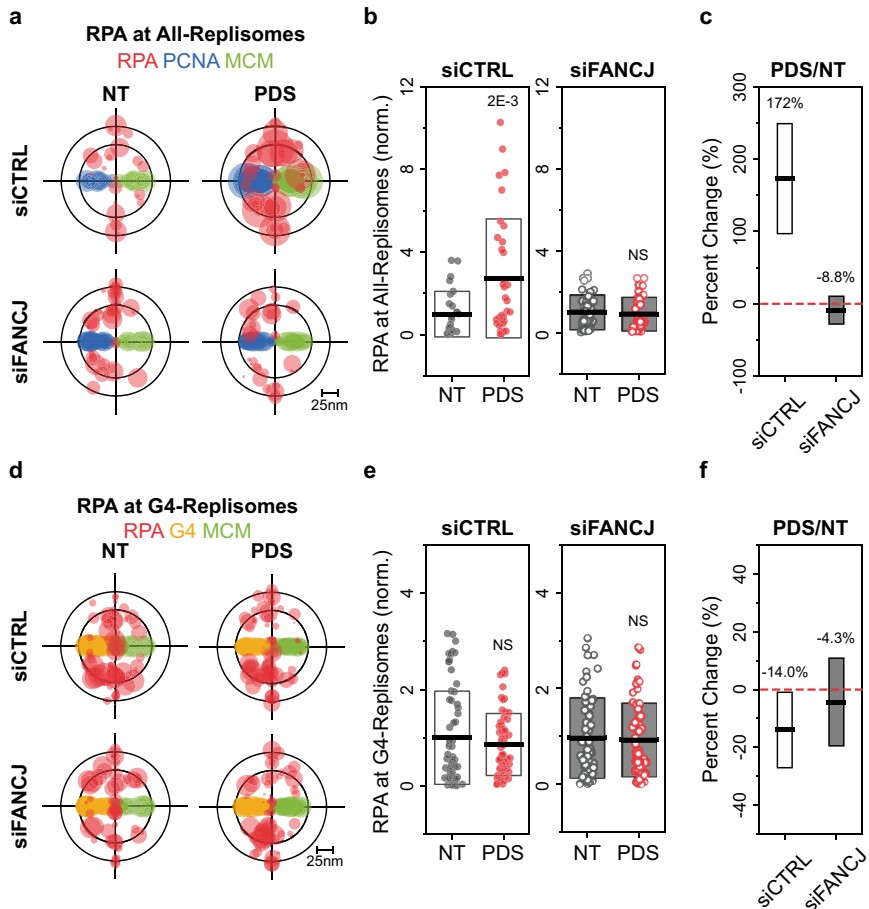

**Fig. 4 Deregulated G4-Replisomes resist RPA loading. a, d** Overlaid TC triplets of RPA, PCNA, MCM (**a**) or RPA, G4, MCM (**d**) from multiple NT or 4 h, 20 μM PDS-treated siCTRL or siFANCJ cells. Circle size of each TC triplet represents the local density of RPA from a given nucleus. The TC triplets are aligned onto the same PCNA-MCM (**a**) or G4-MCM (**d**) plane to define the positions of RPA relative to the replisome complex. **b, e** Local densities of RPA at All-Replisomes (**b**) or G4-Replisomes (**e**) in NT or PDS-treated siCTRL or siFANCJ cells. Individual data points represent result from a single nucleus. Black horizontal line and box height indicate mean ± SD. Values on graph indicate p-values of unpaired two-sample t-tests between NT and PDS-treated cells. **c, f** Percent change in the densities of RPA at All-Replisomes (**c**) or G4-Replisomes (**f**) in siCTRL or siFANCJ PDS-treated compared to NT cells. Values on the graph and black horizontal line represent the respective percent changes, box height indicates the propagated s.e.m. For all experiments, number of cells analyzed and TC triplets identified are listed in Supplementary Table 1.

of G4 L3 (Supplementary Fig. 4b) since the ease with which RPA unfolds less stable G4 constructs partly masks the action of FANCJ and illustrates a redundancy between these two proteins under these conditions. Collectively, these data reveal that the resolution of thermodynamically stable G4 is mediated by an interplay between RPA and FANCJ, wherein FANCJ helicase activity increases the frequency with which stable G4 structures are destabilized, facilitating the loading of RPA to persistently unfold the secondary structure.

**Consequences of RF-coupled G4 accumulation.** Since RPA binding and accumulation at forks during replication stress is an essential step in the initiation of the replication stress response[46], we sought to investigate how the limited binding of RPA at stable RF-coupled G4s affects RPA-mediated signaling. To determine this, we used Western blotting to measure the changes in the accumulation of whole-cell RPA2 phosphorylation (pRPA) signal, which is a marker for ATR-mediated replication stress response. We determined pRPA in siCTRL and siFANCJ U2OS cells following PDS treatment. We found that while PDS-induced stabilization of G4s in siCTRL cells led to a noticeable increase in pRPA signal, PDS treatment of siFANCJ cells showed no significant change in pRPA (Supplementary Fig. 5a). These findings

correspond to the trends in Fig. 4, whereby the obstruction in RPA recruitment to stable RF-coupled G4s would inhibit replication stress response at those sites.

Without the activation of the replication stress response, we hypothesized that persistent stalling of replication forks by G4 stabilization would generate double-strand breaks (DSBs) at the stalled forks[46]. We therefore probed for the DNA damage marker γH2AX in U2OS cells via conventional epifluorescence microscopy and noted a clear induction of γH2AX foci in siFANCJ cells compared to siCTRL cells following G4 stabilization (Supplementary Fig. 5b). Next, we utilized SMLM-TC analysis to determine whether this damage is indeed associated with the persistent G4s accumulating at forks, by measuring the relative abundance of γH2AX signals locally at G4-Replisomes (γH2AX/MCM/G4). Our data revealed an increased accumulation of γH2AX at G4-replisomes in siFANCJ cells upon PDS exposure, while the abundance of the damage signal at G4-Replisomes in siCTRL cells remained unchanged (Fig. 6a–c, Supplementary Fig. 5c). We also examined the densities of γH2AX at All-Replisomes (γH2AX/MCM/PCNA), which showed insignificant changes upon G4 stabilization in either siCTRL and siFANCJ cells (Supplementary Fig. 5d, e). We reasoned that the γH2AX signals we observe at G4-Replisomes arise from few events that are specifically localized at G4-Replisomes and therefore would be averaged out over

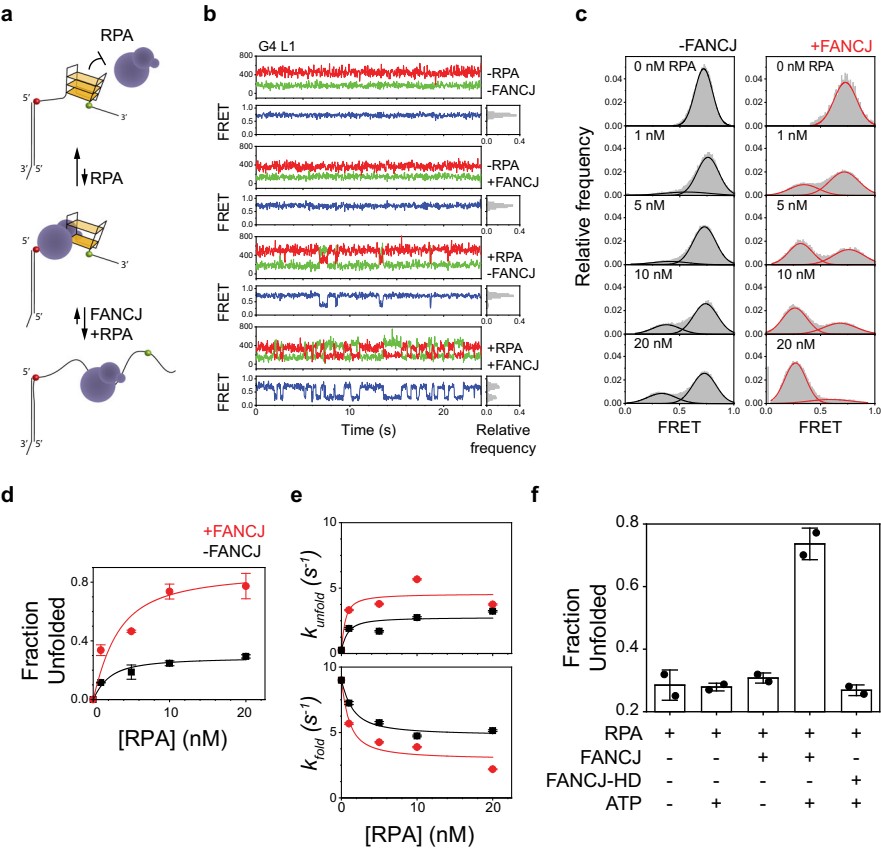

**Fig. 5 Interplay between FANCJ and RPA on G4-resolution. a** Illustration of the FANCJ/RPA-mediated destabilization of a single G4 substrate monitored by smFRET. **b** Representative single-molecule trajectories of the L1 G4 substrates in the presence or absence of 10 nM RPA and/or 100 pM FANCJ+ATP. **c** FRET histograms of L1 G4 in the presence of indicated concentrations of RPA with either no (left panel) or 100 pM FANCJ (right panel). **d** Fractions of unfolded L1 G4 monitored by smFRET as a function of RPA concentration with (red) or without (black) the addition of FANCJ. Error bars represent mean ± s.e.m. **e** Unfolding (top) and folding (bottom) rates for L1 G4 as a function of RPA concentration with (red) or without (black) the addition of FANCJ. Error bar represents the SD of exponential fit. **f** Fraction of unfolded L1 G4 in indicated conditions monitored by smFRET. Data collected from two experiments, shown as black datapoints, were pooled together for analysis. Error bars represent mean ± SD. For all smFRET experiments, a minimum of 100 smFRET trajectories from two independent experiments were used for analysis.

the entire, All-Replisome, population. We concluded that failure in the timely resolution of G4s in siFANCJ cells results in a significant deficiency in stimulating the replication stress response and therefore progresses to DSB generation at G4-associated replisomes.

## Discussion

Mapping the spatial organization and activity of replisomes and associated proteins in cells is crucial for understanding the various obstructions encountered during replication fork progression and for establishing the molecular mechanisms by which such faults are resolved. Repetitive G-rich DNA sequences that can form G4 structures exist throughout our genome and have stirred much interest as to whether these structures spontaneously form in chromosomes and how their formation might influence genomic processes, particularly DNA replication. The interaction of replication machinery with G4s has been the subject of multiple studies, providing important information into the consequences of deregulated and unresolved G4s, and their impact on genomic integrity[15,27]. Nevertheless, much of our knowledge of these encounters is based on studies of replication at specific genomic loci that are known to contain G4s, whereas the RF-coupled formation of G4s, which was hypothesized to frequently and spontaneously occur as replication forks progress through the multiple G4-forming motifs that are widely distributed throughout the genome, have not been addressed. This is because

methodologies for monitoring the transient occurrence of such structures, as well as their association with, and influence on, replication machinery at high resolution in intact cells has not been realized.

Our multi-color SMLM platform coupled with TC image data-mining approach[28,29] provide nanoscale quantitative mapping of the spatial-association between endogenous replisome complexes and G4s within intact single cells. Using this approach, we directly visualized the RF-coupled formation of G4 structures and have defined the spatial configurations of these events. We find that these G4s are predominantly positioned between the MCM helicase and nascent DNA, indicating that G4s form at newly unwound ssDNA prior to nascent DNA synthesis. The formation of G4s within replisomes locally obstruct replication fork progression and impede RPA ssDNA protection. We further show that mild inhibition of the replicative polymerases by APH, which induces helicase-polymerase uncoupling and ssDNA exposure[52], increases the frequency of RF-coupled formation of G4s. Importantly, G4-Replisomes induced by APH treatment exhibit the same distinct behavior as the G4-Replisomes observed in untreated cells. We conclude that transient, RF-coupled G4 formation that occurs during normal replication is likely to cause brief fork uncoupling events, and therefore requires continuous and timely regulation.

Of particular significance is our observation that the formation of G4-Replisomes precludes the binding of RPA. MCM

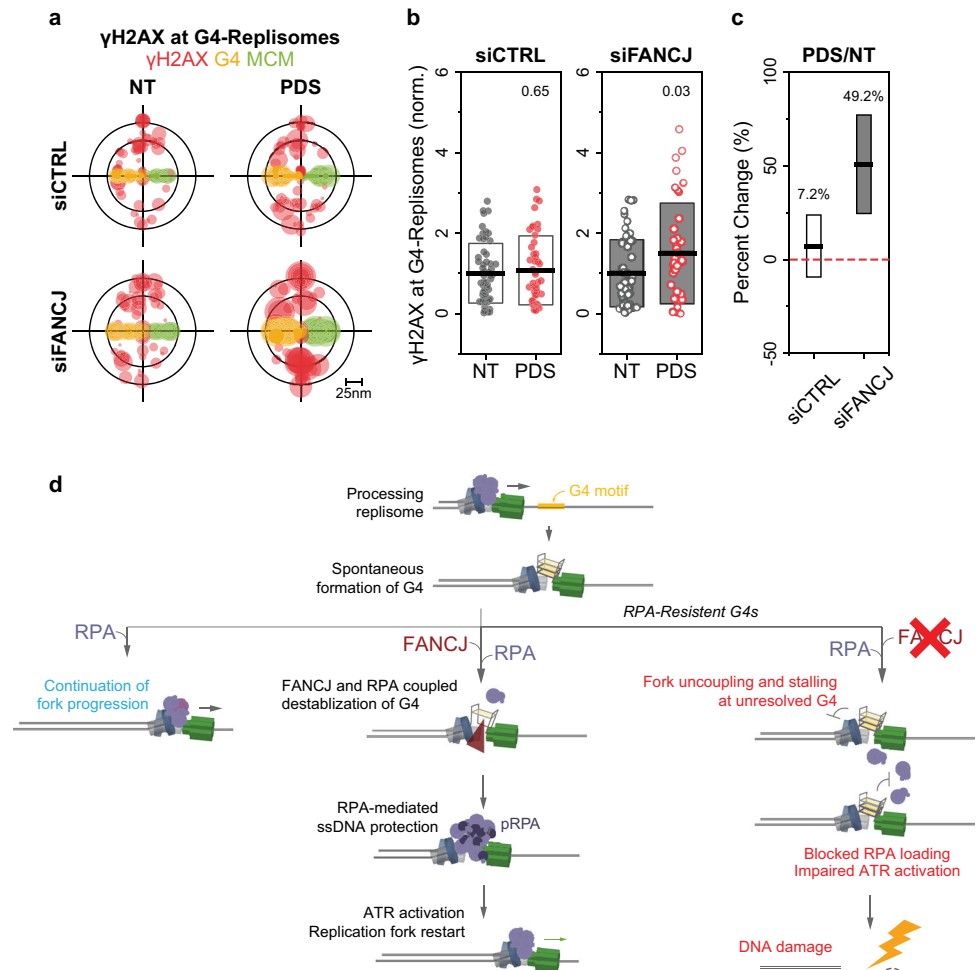

**Fig. 6 Unresolved G4-Replisome accumulation causes DNA damage. a** Overlaid TC triplets of γH2AX, G4, MCM from multiple NT or 4 h, 20 μM PDS-treated siCTRL or siFANCJ cells. Circle size of each TC triplet represents the local density of γH2AX from a given nucleus. The TC triplets are aligned onto the same G4-MCM plane to define the positions of γH2AX relative to the replisome complex. **b** Local densities of γH2AX at G4-Replisomes in NT or PDS-treated siCTRL or siFANCJ cells. Individual data points represent result from a single nucleus. Black horizontal line and box height indicate mean ± SD. Values on graph indicate *p*-values of unpaired two-sample *t*-tests between NT and PDS-treated cells. **c** Percent change in the densities of γH2AX at G4-Replisomes in siCTRL or siFANCJ PDS-treated compared to NT cells. Values on the graph and black horizontal line represent the respective percent changes, box height indicates the propagated s.e.m. For all experiments, number of cells analyzed and TC triplets identified are listed in Supplementary Table 1. **d** Model of FANCJ/RPA-mediated G4 regulation and consequences at replicating forks: during DNA replication at G4 motifs, the exposed ssDNA stretches may fold into G4 structures. RPA can destabilize some G4s, allowing for seamless continuation of replication fork progression. For G4s with higher thermostability (RPA-Resistant G4s), RPA alone would not be sufficient to destabilize them, therefore requiring facilitation by FANCJ, whose initial unwinding of G4 facilitates the loading of RPA onto ssDNA, thereby activating a proper replication stress response to promote replication fork restart. Accordingly, loss of FANCJ results in accumulation of RPA-resistant G4s at forks, leading to persistent fork uncoupling and stalling, the silencing of replication stress signaling, eventually leading to DNA damage accumulation at G4-Replisomes.

unwinding of parental DNA during replication exposes ssDNA especially on the lagging strand, which is rapidly coated and protected by RPA[15]. Although our approach cannot distinguish between G4s that form at the leading strand and those at the lagging strand, we found that RF-coupled formation of G4s consistently exhibit resistance to RPA recruitment, even upon APH-induced fork uncoupling. Previous in vitro studies have shown that RPA could bind and unfold G4 structures[48,49], suggesting that RPA could play a direct role in counteracting these structures during replication. However, a study using smFRET assays has systematically characterized the binding and unfolding ability of RPA on diverse G4 structures, and showed that the more thermodynamically stable G4s, including the ones with shorter ssDNA loop lengths, are remarkably resistant to RPA binding in vitro[50]. Intriguingly, it was recently shown that G4 motifs with loop size of 1-nucleotide are significantly prevalent in

the human genome[51] and that they contribute to genomic instability[53]. We therefore hypothesized that in cells the more stable RF-coupled G4s cannot be unfolded by RPA alone, and thus need to be resolved by additional factors in order to maintain normal replication fork progression.

A growing list of human helicases, including FANCJ, BLM, WRN, PIF1, and RTEL1 have been shown to have G4-specific activities, with the abilities to unwind G4 DNA structures in vitro, along with roles in maintaining genome integrity[15]. The FANCJ helicase[38], whose mutations are associated with a rare subtype of Fanconi anemia[54,55] as well as early-onset breast and ovarian cancers[56], have been proposed to participate in the maintenance of replication integrity. Several studies have outlined the importance of FANCJ in G4 regulation, presumably during replication, in model organisms, with deficiencies in FANCJ homologs leading to hypersensitivity to G4-stabilizing ligands, resulting in

reduced replication[39,40], impaired cell proliferation, DNA damage induction, and elevated apoptosis[20]. Here, we provide direct evidence pertaining to the role of FANCJ in regulating RF-coupled formation of G4 structures. Our data revealed an increased frequency in stable G4s at replication forks upon FANCJ depletion or inactivation, which prevents RPA from loading onto these sites, leading to DNA damage that is specifically localized at G4-Replisomes. Importantly, the suppression of RPA binding at the damaged G4-Replisomes also corresponds to defective replication fork signaling via pRPA, in agreement with previous studies showing that loss of FANCJ inhibits RPA-ATR signaling[57,58], and that the inability of RPA to bind repetitive DNA sequences can suppress ATR checkpoint control[46,59–61]. The close cooperation between RPA and FANCJ in resolving stable G4s is further realized via our in vitro smFRET measurements, which show that FANCJ helicase activity is required for loading RPA onto stable short-loop G4s, which are otherwise resistant to RPA binding. These results also offer insights into previous studies wherein RPA was shown to stimulate the unwinding activity of FANCJ[20,47]. Given the variety of specialized G4 helicases and the known associations amongst some of them[19], as well as with RPA[62,63], it is likely that there is at least a partial redundancy in the roles of these helicases in assisting the RPA-mediated resolution of RF-coupled G4 formation like that observed for FANCJ. Indeed, a recent study in the *Saccharomyces cerevisiae* model system demonstrated that replication through G-rich minisatellites is enabled via a cooperation between RPA and the G4 helicase Pif1[63]. Further studies into the mechanisms governing RF-coupled G4 resolution in human cells are needed to clarify the specific contribution of different factors to these pathways.

Together, our findings demonstrate that DNA G4 structures normally form at a subset of replisomes upon unwinding of parental DNA, altering their replication behavior. In the case of perturbed replication, when G4s formed at replication forks are not properly resolved, these persistent G4s could lead to genomic instability. Our study provides novel mechanistic insights whereby the resolution of stable RF-coupled G4s depends on a collaboration between RPA and FANCJ, which also facilitates proper signaling and related replication stress response at these replisomes. We propose the following model, as illustrated in Fig. 6d: RF-coupled G4 formation induces local and temporary helicase-polymerase uncoupling events. Generally, replication fork uncoupling will result in RPA accumulation, RPA-mediated ATR activation and replication stress response[46]. While RPA can readily bind and destabilize some G4s, the more stable G4s cannot be unfolded by RPA alone and therefore will be inhibitory to RPA-mediated signaling. These G4s need to be destabilized by the FANCJ helicase to facilitate the subsequent loading of RPA, thereby maintaining DNA synthesis and proficient ATR replication stress response at these forks. Accordingly, loss of FANCJ results in a failure of timely removal of RPA-resistant, stable G4s at forks, leading to DNA damage accumulation along with defective replication stress signaling. The combination of persistent DNA lesions and suppressed signaling would have major mutagenic consequences that could explain why G4 motifs are frequently associated with genomic rearrangements in cancer genomes[64,65].

## Methods

**Cell culture and drug treatments.** U2OS cells (ATCC HTB-96) and Human cells Human cervix epithelioid carcinoma Flp-In T-REx (HeLa FIT) cells[44,66] were cultured in Dulbecco's modified Eagle's medium (DMEM) (ThermoFisher 11965) with 10% fetal bovine serum (Gemini Bio., 100-106) and 1% Pen-Strep (ThermoFisher, 15140) inside a 37°C incubator at a 5% $CO_2$-containing atmosphere. For all imaging experiments, cells were trypsinized and seeded on glass coverslips (Fisher Scientific, 12-548-B) in six-well plates at low density and allowed to attach.

Drug treatments and/or siRNA transfection were performed directly on cells on coverslips.

To investigate the effects of aphidicolin on G4 formation and association with the replisome in S-phase cells, fully attached cells were arrested to G0/G1 phase using serum starvation for 72 h. Cells were subsequently released in complete medium for a further 16 h to produce a predominantly early/mid-S phase cell population. Cells were then treated with different concentrations of APH (Abcam, 142400) for 1 h before analyses. The specific concentrations of APH are indicated accordingly in the text or in the figure legends.

To investigate the mechanisms of G4 regulation in cells, siRNA transfected cells were treated with 20 μM PDS (Sigma, SML0678) for 1, 4, or 24 h before analyses. The specific durations of PDS treatment are indicated accordingly in the text and/or in the figure legends. We emphasize that the addition of PDS only increases the melting temperature (thermostability) of already folded native G4s[66–68], resulting in a slight enrichment in the frequency of stable native G4s at forks, and provided improved in situ probing of their presence, effects, and regulations on replication forks.

**FANCJ knockout.** siRNAs reverse transfections were performed using Lipofectamine RNAiMax (ThermoFisher) following the manufacturer's instructions. SMLM experiments were done 72 h after transfection. Knockdown efficiency was confirmed via Western blot analysis (Fig. S8). The siRNAs used in this study are:
FANCJ: 5′-TAGATAGTATGGTCAACAATA-3′ (QIAGEN, Hs_BRIP1_6, SI03110723)
CONTROL: 5′-AATTCTCCGAACGTGTCACGT-3′ (QIAGEN 1027310)
Details of CRISPR-mediated FANCJ knockout (FANCJ-KO) HeLa cell-line with or without wild-type FANCJ (FANCJ-WT) or a helicase-dead FANCJ$^{K52R}$ (FANCJ-HD) complementation were described previously[44].

**Permeabilization and fixation.** An optimized permeabilization and fixation protocol were used to remove the majority of the cytoplasm and non-chromatin bound proteins in order to minimize nonspecific antibody labeling, which could significantly contribute to the noise for image analysis. Cells were permeabilized with 0.5% Triton X-100 in ice-cold CSK buffer (10 mM Hepes, 300 mM Sucrose, 100 mM NaCl, 3 mM $MgCl_2$, pH = 7.4) for 10 min at room temperature. Following pre-extraction, cells were washed once with PBS, then fixed in 3.7% paraformaldehyde (Electron Microscopy Sciences, 15714) in PBS for 30 min at room temperature. Cells were then washed twice with PBS and blocked with blocking buffer (2% glycine, 2% BSA, 0.2% gelatin, and 50 mM $NH_4Cl$ in PBS) at least overnight at 4°C prior to immunofluorescence staining and imaging.

**Immunofluorescence labeling.** For nascent DNA detection, cells were treated with 10 μM EdU for 15 min before fixation, so that EdU only incorporates into newly synthesized DNA in S-phase cells through endogenous replication. Incorporated EdU was labeled using Click-iT™ Plus EdU Alexa Fluor™ 647 Imaging Kit (ThermoFisher, C10640) after fixation. DNA G4, MCM, PCNA, RPA, and γH2AX were labeled either directly by Alexa Fluor-conjugated primary antibodies in blocking buffer for 1 h, or indirectly using primary antibodies for 1 h, then Alexa Fluor secondary antibodies for 30 min. All staining steps were done at room temperature. All antibodies used in IF studies are listed in Supplementary Table 2.

**SMLM imaging.** After immunofluorescence staining, coverslips with fixed cells were mounted on microscope glass slides with freshly prepared SR imaging buffer (1 mg/mL glucose oxidase (Sigma, G2133), 0.02 mg/mL catalase (Sigma, C3155), 10% glucose (Sigma, G8270), 100 mM mercaptoethylamine (Fisher Scientific, BP2664100) in PBS, pH = 8) flowed through.

All raw SMLM-SR images were acquired using a custom-built optical imaging platform based on a Leica DMI 300 inverse microscope. 750 nm (UltraLaser, MDL-III-750-500), 639 nm (UltraLaser, MRL-FN-639-800), 561 nm (Cobolt), 488 (OBIS) laser lines were adjusted to 1.5, 0.8, 1.0, 0.8 kW/cm², respectively. The laser lines were combined using appropriate dichroic and focused onto the back aperture of an HCX PL APO 63X NA = 1.47 OIL CORR TIRF (Zeiss) Objective via a multi-band dichroic (FF408/504/581/667/762-Di01). To increase power density and limit out-of-plane fluorescence, a Highly Inclined and Laminated Optical (HILO) illumination configuration was achieved by translating the excitation beam laterally across the back aperture of the objective. Fluorescence emission was expanded with a 2X lens tube, corrected by a chromatic aberration correction lens (Thorlabs, AC254-300-A), and was collected on a sCMOS camera (Photometrics, Prime 95B). Fluorescence signals were collected sequentially using the corresponding single-band pass filters in a filter wheel (ThorLabs, FW102C): AF750 (Semrock, FF02-809/81), AF488 (Semrock, FF01-531/40), AF647 (Semrock, FF01-676/37), AF568 (Semrock, FF01-607/36). A 405 nm laser line (MDL-III-405-150, CNI) was introduced to enhance recovery of dark state fluorophores when required. 2000 Frames at 33 Hz were acquired for each color. Image acquisition was done using the Micro-Manager (v1.4) software.

**Mapping and alignment of images from different colors.** Mapping among different channels for multi-color imaging was carried out using a polynomial morph-type mapping algorithm in order to correct the chromatic aberrations

caused by the varying diffraction behaviors of different wavelength emissions[29]. Before each experiment, a calibration map was generated by imaging spatially separated fluorescent beads (ThermoFisher, T-7279) in each of the four channels. A 2nd polynomial function was optimized to fit the localizations of the beads in each of the AF750, AF568, and AF488 channels to their locations in the AF647 channel. This optimized 2nd polynomial function is then used to map the molecular localizations of the experimental samples in each of AF750, AF568, AF488 channels to the AF647 channel.

**Single-molecule localization.** Each frame of the raw image stack was firstly box-filtered with a box size of 4 times of the FWHM of a 2D Gaussian PSF. Note that each pixel of the image was weighted by the inverse of its pre-calibrated variance during the box-filtering[69]. The low-pass filtered image was then extracted from the raw image for rough local maxima recognition and localization. All the $7 \times 7$ pixel regions around all the local maxima from all frames of the image stack were then submitted for 2D-Gaussian multi-PSF fitting[70], which is performed by GPU (Nvidia GTX 1060, CUDA 8.0) using the Maximum Likelihood Estimation (MLE) algorithm. In brief, the likelihood function of each pixel was built by the convolution of (1) the Poisson distribution of the shot noise from the photons emitted from fluorophores nearby and (2) the gaussian distribution of the inherent readout noise of each pixel pre-calibrated as mentioned above. The fitting accuracy was then estimated by Cramér-Rao lower bound (CRLB), and the distribution of the accuracy of all sequential localizations were fitted into a skew-Gaussian distribution. Any localizations appearing in consecutive frames within 2.5 times of the localization precision were considered as one blinking event. Such localizations were weighted by the inverse of its own CRLB determined variance and averaged into one localization in order to minimize overcounting during Auto-PC computation. For display purpose, the representative images were generated by rendering the raw coordinates into 10 nm pixel canvas and convolved with a 2D-Gaussian ($\sigma = 10$ nm) kernel.

**Triple-correlation function.** Details of the TC algorithm were described previously[28,29]. Briefly, the TC Function is defined as Eq. (1),

$$f(\mathbf{r_1}, \mathbf{r_2}) = \frac{\langle \delta\rho_1(\mathbf{R})\delta\rho_2(\mathbf{R}+\mathbf{r_1})\delta\rho_3(\mathbf{R}+\mathbf{r_2}) \rangle_{\mathbf{R}}}{\langle \rho_1(\mathbf{R}) \rangle_{\mathbf{R}} \langle \rho_2(\mathbf{R}) \rangle_{\mathbf{R}} \langle \rho_3(\mathbf{R}) \rangle_{\mathbf{R}}} \quad (1)$$

where $\langle \rho_i(\mathbf{R}) \rangle_{\mathbf{R}}$ denotes the average density of the detections from the $i$th of the three-color channels within the Region-Of-Interests (ROI, a ~$6 \times 6$ μm² square at the center of the 3-color SMLM image of a nucleus) and $\delta\rho_i(\mathbf{R}) = \rho_i(\mathbf{R}) - \langle \rho_i(\mathbf{R}) \rangle_{\mathbf{R}}$ denotes the local density fluctuation at $\mathbf{R}$. The implementation of the TC to define any significant TC triplets is illustrated in Supplementary Note 3.

**Estimation of the local density within a TC triplet pattern via TC Function.** $\langle \delta\rho_1(\mathbf{R})\delta\rho_2(\mathbf{R}+\mathbf{r_1})\delta\rho_3(\mathbf{R}+\mathbf{r_2}) \rangle_{\mathbf{R}}$ defines, on average, the product of the local density of the three species within a triplet pattern $\triangle(\mathbf{r_1}, \mathbf{r_2})$, while $\langle \delta\rho_1(\mathbf{R})\delta\rho_2(\mathbf{R}+\mathbf{r_1}) \rangle_{\mathbf{R}}$ stands for the average product of the two species correlating at $\mathbf{r_1}$. Similar to the conditional probability, the local density of the third species within the triple-pattern is therefore estimated as the 'conditional' local density at $\mathbf{r_2} - \mathbf{r_1}$ given a pair correlating at $\mathbf{r_1}$ (2):

$$C_3(\mathbf{r_1}, \mathbf{r_2}) = \frac{\langle \delta\rho_1(\mathbf{R})\delta\rho_2(\mathbf{R}+\mathbf{r_1})\delta\rho_3(\mathbf{R}+\mathbf{r_2}) \rangle_{\mathbf{R}}}{\langle \delta\rho_1(\mathbf{R})\delta\rho_2(\mathbf{R}+\mathbf{r_1}) \rangle_{\mathbf{R}}} \quad (2)$$

Such local density within a triplet pattern is validated via simulations in Supplementary Note 3.

**Computation of TC.** Since SMLM data consists of coordinates other than intensity values at each pixel across the entire image canvas, we directly calculated the TC as its definition (1) by visiting each coordinate in the first channel, and calculated $\delta\rho_2(\mathbf{r_1})$ and $\delta\rho_3(\mathbf{r_2})$ in the second and third channels at $\mathbf{r_1}$, and $\mathbf{r_2}$ displaced from the visited coordinate, respectively. Moreover, since the triplets are randomly oriented in the ROI, the TCF at $\mathbf{r_1} = (r_1, \theta), \mathbf{r_2} = (r_2, \theta + \theta)$ was averaged along $\theta \in [-\pi, \pi]$, and $f(\mathbf{r_1}, \mathbf{r_2})$ was thus transformed to $f(r_1, r_2, r_3)$ where $r_3^2 = r_1^2 + r_2^2 + 2r_1r_2\cos\triangle\theta$.

**Auto-Correlation (AC).** For AC analyses, a ~$6 \times 6$ μm² square at the center of each SMLM imaged nucleus were cropped and submitted to the Auto-PC function (3).

$$g(\mathbf{r}) = \frac{\langle \delta\rho(\mathbf{R})\delta\rho(\mathbf{R}+\mathbf{r}) \rangle_{\mathbf{R}}}{\langle \rho(\mathbf{R}) \rangle_{\mathbf{R}}^2} \quad (3)$$

Note that artificial blinking events were eliminated by averaging the localizations appearing in consecutive frames within 2.5 times of the localization precision as discussed above. The output correlation profile $g(r)$ was plotted as the function of

pair-wise distances $r$, and fitted into a two Gaussian model as Eq. (4):

$$g(r) = \frac{1}{4\pi\sigma^2\langle\rho\rangle}\exp\left(-\frac{r^2}{4\sigma^2}\right) + A\exp\left[-\frac{r^2}{4(\sigma^2+r_{app}^2)}\right] + 1 \quad (4)$$

where $\sigma$ and $\langle\rho\rangle$ denotes the localization precision and the averaged density of the examined protein within the $6 \times 6$ μm² square (termed as the 'global density'), respectively. Each focus was modeled as a Gaussian distribution and the second term in Eq. (4) is its auto-correlation form that convoluted with the stochastic sampling (the first term). $A$ is proportional to the average probability of finding molecules around each other and $r_{app}$ stands for the average sigma radius of the Gaussian modeled focus. The averaged molecular content $\langle N \rangle$ of each focus within such $6 \times 6$ μm² square was then calculated as Eq. (5).

$$\langle N \rangle = \iint_{-}^{+}\langle\rho\rangle A\exp\left(-\frac{x^2+y^2}{2r_{app}^2}\right)dxdy = 2\pi\langle\rho\rangle A r_{app}^2 \quad (5)$$

**Recombinant protein expression and purification.** FANCJ expression and purification were described previously[44]. In brief, Sf9 cells were infected with recombinant baculoviruses encoding for N-terminally Flag-tagged FANCJ WT or FANCJ HD. 48 h after infection, the cells were spun down and the pellet was lysed for 1 h in buffer A (50 mM Na$_2$HPO$_4$/NaH$_2$PO$_4$ (pH = 7.4), 150 mM NaCl, 10% glycerol, 0.01% NP-40, 0.5 mM EDTA, 1% Triton X-100) supplemented with protease inhibitors (Roche). Lysed cells were spun down and the supernatant was incubated on Flag M2 beads (Sigma-Aldrich) for 2 h at 4°C. Subsequently, the beads were washed twice with buffer B (50 mM Na$_2$HPO$_4$/NaH$_2$PO$_4$ (pH = 7.4), 150 mM NaCl, 10% glycerol, 0.01% NP-40, 0.5 mM EDTA), followed by one wash with buffer A, and one wash with buffer C (50 mM Na$_2$HPO$_4$/NaH$_2$PO$_4$ (pH = 7.4), 150 mM NaCl, 10% glycerol, 0.01% NP-40, 0.5 mM EDTA, 1%, 5 mM MgCl$_2$,) supplemented with 5 mM ATP. Finally, the beads were washed extensively in buffer B and eluted for 1 h in buffer B containing 200 μg/ml 3× Flag peptide (Sigma-Aldrich).

RPA expression and purification were described previously[71].

**smFRET assays.** To observe G4 unfolding using smFRET, a PEG-coated imaging surface with 50 pM DNA immobilized via biotin–Neutravidin linkage was prepared. Reaction buffer containing RPA and/or 100 pM FANCJ is then prepared at room temperature in a buffer composed of 20 mM pH = 7.5 TrisAc, 50 mM KAc, 10 mM MgAc, 0.8% (w/v) glucose, 0.5 mg/mL glucose oxidase, 0.4 μg/mL catalase, 5 mM Trolox, 1 mg/mL BSA, 1 mM ATP, and 2 mM DTT. Concentrations of RPA and/or FANCJ proteins varies and are indicated accordingly in the text or in the figure legends. The reaction was immediately flowed into the imaging chamber right before imaging acquisitions. All DNA substrates used in smFRET studies are listed in Supplementary Table 3.

Single-molecule imaging was performed on a custom-built Total-Internal Reflection Fluorescence Microscopy (TIRFM) system based on a modified inverted microscope (IX70, Olmpus) equipped with a high NA TIRF objective (PLAN APO; 100×; NA, 1.45; OIL TIRF, Olympus) as previously described[72]. Briefly, the microscope was coupled to 532- and 640-nm solid-state lasers to excite the sample at TIRF illumination mode for improved signal-to-noise ratio and to reject out-of-plane fluorescence. Sample emission was collected and split into two channels through a dichroic (FF660, Semrock) and emission narrow-band bandpass filters (HQ580/60 and ET690/50; Chroma) in conjunction with the use of an Optosplit II (Cairn Research) to image two colors simultaneously onto a single EMCCD camera (Andor iXon3). Movies consisting of 800 frames were acquired for analyses with each frame having an exposure of 30 ms.

smFRET trajectory analyses were performed in Matlab. Briefly, molecules were identified using custom-written-mapping routines to obtain intensity vs time trajectories, followed by idealization of the trajectories and measurement of dwell times using ebFRET[73]. FRET efficiencies were approximated as the ratio between the acceptor intensity and the sum of acceptor and donor intensities. Each smFRET histogram was generated by a minimum of 100 trajectories from two independent experiments. The peaks corresponding to the folded and unfolded populations were fitted to two independent gaussian curves, and the area from each curve was used to calculate the unfolded fraction. The measured dwell times were fitted to single exponential decays to calculate folding ($k_{fold}$) and unfolding ($k_{unfold}$) values. Data collected from two independent experiments were pooled together for analysis.

**SiMPull assays, imaging, and analysis.** To validate that the G4 antibody 1H6[23] can indeed bind on DNA G4, a modified SiMPull assay was performed[74]. Biotin-tagged anti-mouse or anti-rabbit IgG was first immobilized to a PEG-coated quartz surface with biotin-neutravidin linkage. 1H6 antibody was then flowed into the imaging chamber, followed by the flow through of either Cy3/Cy5-labeled DNA G4 or T20 single-stranded DNA oligonucleotides. The chamber was then washed multiple times. The presence of Cy3/Cy5 signals, which indicates the capture of the DNA oligos by the antibodies, was observed and imaged using the same microscope system as smFRET experiment described above. The amount of captured

oligos was then counted using ImageJ "Analyze Particles" plugin. The experiment and washes were carried out at room temperature in a buffer composed of 50 mM Tris-HCl (pH = 8.0), 2 mM MgCl₂, 100 mM KCl and an oxygen scavenging system (1 mg mL⁻¹ glucose oxidase, 0.4% (w/v) D-glucose, 0.02 mg mL⁻¹ catalase and 2 mM Trolox). Detailed information for antibodies and DNA substrates used are listed in Supplementary Tables 2 and 3.

**Colocalization analyses**. To visualize the degree of colocalization between the two G4 antibodies 1H6[23] and BG4[22], each nucleus was manually outlined to generate an ROI for independent analysis. An automatic Otsu threshold[75] was then applied and the clusters defined for each color for each nucleus. The extent of overlaps between the two colors were then visualized using ImageJ "AND" function. To generate a baseline of expected random colocalization, the clusters of one color were redistributed within the ROI using a Monte Carlo randomization algorithm[76] and the extent of random overlaps between the two colors were visualized using the same ImageJ function. The abundance of the overlapped foci indicates significant, non-random colocalization between the two antibodies.

To further quantify the level of colocalization, we calculated the cross-correlation (Eq. (6)) between 1H6 and BG4 in both experimental data and randomized data (See Fig. S3 and Fig. S4 in Chen, Y. H. et al.[30] for randomization procedure).

$$c(\mathbf{r}) = \frac{\langle \delta\rho_1(\mathbf{R})\delta\rho_2(\mathbf{R+r})\rangle_{\mathbf{R}}}{\langle \rho_1(\mathbf{R})\rangle_{\mathbf{R}}\langle \rho_2(\mathbf{R})\rangle_{\mathbf{R}}} \qquad (6)$$

In brief, the 1H6 and BG4 signal from the ROI of the same nucleus were submitted for co-localization test via cross-correlation, whilst the cross-correlation between 1H6 and BG4 signal from the ROIs of different nuclei served as the correlation level for two distributions that random to each other.

**Fraction analysis**. To determine the colocalization between PCNA and G4 foci detected in SMLM, the single-molecule localizations of both species were first submitted to home-written Density-Based Spatial Clustering of Applications with Noise (DBSCAN) algorithm for foci segmentation. The number of minimum points that could form a DBSCAN focus was set at 3 and the threshold distance was set at 15 nm. After DBSCAN segmentation, the Nearest-Neighboring Distance (NND) (edge-to-edge) between the two species were calculated and clusters that maintain an NND < = 5 nm were designated as a pair of co-localized clusters. The fraction was then calculated by dividing the number of clusters of either of the species from the number of co-localized pairs. To ensure that the detected colocalizations are non-random, the levels of colocalization were compared to random distributions, which is generated via randomly repositioning and orienting the clusters.

**γH2AX imaging**. To detect γH2AX signals, S-phase U2OS cells were pulsed-labeled with EdU and immuno-stained with anti-γH2AX antibody (Supplementary Table 2). After staining coverslips were mounted onto glass slides using VEC-TASHIELD Antifade Mounting Medium with DAPI (VectorLabs, H-1200) and imaged with a Keyence BZ-X800 microscope using the Keyence BZ-X software (Keyence). At least 300 EdU-positive cells were acquired.

**Western blotting**. Cells were harvested using Laemmli sample buffer (Bio-Rad) containing (final concentration) 2% SDS, 10% glycerol, 5% 2-mercaptoethanol, 0.002% bromphenol blue, and 60 mM Tris-HCl (pH ~ 6.8) and lysed by heating the samples at 95°C for 15 min. Protein extracts were calibrated and resolved by SDS-polyacrylamide gel electrophoresis on Nupage 4–12% Bis-Tris, 3–8% Tris-Acetate gels (Invitrogen), or 4–15% TGX gels (Bio-Rad) in 1 × Tris-Glycine-SDS buffer. Proteins were transferred onto polyvinylidene difluoride (PVDF) membrane (Millipore) and incubated in 5% milk in TBST for 1 h at room temperature. The membrane was then incubated with primary antibodies overnight at 4°C, followed by incubation with secondary antibodies conjugated with horseradish peroxidase for 1 h at room temperature. Blots were detected using an Enhanced Chemilu-minescence Detection Kit (GE Healthcare) and were developed with a LICOR Odyssey imager. All antibodies used in WB studies are listed in Supplementary Table 2.

**Reporting summary**. Further information on research design is available in the Nature Research Reporting Summary linked to this article.

## Data availability
All imaging and single-molecule data constitute a sizable dataset (>10TB) that cannot be reasonably maintained online. Raw data will be made available by the corresponding author upon request. Source data are provided with this paper.

## Code availability
Codes for the dTC and dPC algorithms, as well as a testing demo (with simulation codes) are available at https://github.com/yiny02/direct-Triple-Correlation-Algorithm. The code is for Research and Educational Purposes for Non-Profit Academic and/or Research Institutions.

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

## Acknowledgements

This work was supported by NIH grants 1R35GM134947-01 (E.R.), 1R01AI153040-01 (E.R.), 1P01CA247773-01/549 (E.R.), R01ES031658 (T.T.H.), R01CA225018-02 (S.B.C.); American Cancer Society grant RSG-16-241-01-DMC (E.R.); V Foundation BRCA Research grants (E.R. and T.T.H.). W.T.C.L. was supported, in part, by American Heart Association Summer 2018 Predoctoral Fellowship 18PRE33960303. The lab of K.G. was part of COST action CA15133, and has received funding from the Swiss National Science Foundation PP00P3_172959/1 and the Human Frontier Science Program CDA00043/2013-C. M.M. is supported by the French National Research Agency, the French National Cancer Institute and the French National League Against Cancer (équipe labellisée). We thank members of the Rothenberg laboratory for critical discussion.

## Author contributions

W.T.C.L. and E.R. designed the study and wrote the manuscript with input from all the authors. W.T.C.L. performed the imaging and SimPull experiments. W.T.C.L. and Y.Y. analyzed the data using computational pipelines developed by Y.Y. Y.Y. maintained equipment and performed calibration measurements. M.J.M. performed the smFRET experiments. P.T. performed the Western blot experiments supervised by T.T.H. P.G. performed the antibody validation experiment. D.C.O., K.G., S.B.C., and M.M. provided reagents.

## Competing interests

The authors declare no competing interests.
