## [Peer Review File · Nature Communications]

REVIEWER COMMENTS

Reviewer #1 (Remarks to the Author):

This study is a technical tour-de-force providing the first direct proof for the presence of G4-DNA in the replication fork. This major advance has become possible by combining a multi-color single-molecule localization microscopy (SMLM) with unbiased automated pattern recognition protocol employing the Triple-Correlation (TC) function. In brief, nascent DNA in a human cell culture was pulse-labeled by EdU and co-stained with validated antibodies against the MCM-helicase, PCNA, RPA and G4-DNA. The S-phase nuclei were then selected and analyzed by the aforementioned approach. As the result, the authors came to the following conclusions:

(1) Roughly 2% of all replisomes in human cells contained G4-DNA. The fraction of G4-containing replisomes was further increased 1.5-fold when the replicative helicase and DNA polymerases were partially uncoupled by treatment with mild aphidicolin concentrations.

(2) Most importantly, the G4-DNA was located exactly between PCNA/nascent DNA and MCM signals. While this was suspected before, this is the first direct visualization of the G4-DNA location within the replisome. This observation alone warrants the publication in Nat Com.

(3) Another very important observation is that G4-containing replisomes are deprived of RPA even under aphidicolin treatment, as opposed to the regular replisomes. This directly confirms that RPA alone does not efficiently bind to G-rich DNA stretches in ssDNA allowing them to fold into the G4-conformation.

(4) The EdU incorporation within G4-containing replisomes appeared to be 20% lower than that in regular replisomes, supporting the long-suspected notion that once formed, G4-DNA inhibits further DNA synthesis.

(5) Finally, the authors' siRNA experiments has demonstrated that depletion of FANCD1 DNA-helicase results in the enrichment of G4-containing replisomes. Their subsequent in vitro analysis by smFRET showed that FANCD1 unwinds G4-DNA starting from the loops, which in combination with RPA effectively disentangles these structures.

Overall, this is an outstanding study that covers all the bases. Nothing to add or subtract.

Reviewer #2 (Remarks to the Author):

This manuscript from the Rothenberg lab used three color super-resolution microscopy to study the spatial relation between replication machineries (PCNA and MCM as replication fork markers), newly synthesized DNA (EdU), RPA and G4 DNA. Although G4 DNA's relation to DNA replication and the requirement of FANCD1 for resolving G4 DNA have been established, direct evidence of G4 DNA forming as a result of nascent DNA synthesis is lacking, and this study fills the gap. The work shows that the vast majority of replication forks (>97%) do not have an associated G4, but a small fraction does (2.3% vs 1% of random colocalization). They also find evidence that G4 is situated between EdU and MCM/PCNA, suggesting that it forms between the helicase that unwinds the genomic DNA and the replication machinery, which would be consistent with the notion that the CMG helicase complex somehow is able to bypass a G4 structure. They also find that FANCD1 is needed to overcome chemically stabilized G4 and that RPA is recruited to non-G4 associated forks and is phosphorylated, and this effect is FANCD1-dependent. They provide in vitro evidence that a very small amount of FANCD1 is enough to load RPA on a stable G4 structure. Overall, the work is comprehensive and paints a consistent picture, and is a potentially good candidate for Nature Communications.

1. Line 62. direct evidence as to the formation of RF coupled G4s: This is a key point of the manuscript, and needs to be firmly established. Only 2.2% of observed RFs show G4 signal. This is probably not surprising because one would assume only a small fraction of RFs have an unresolved

G4, either because there is no G4 forming sequencing there or because it has already been resolved. However, such a small fraction raises some concerns about the robustness of this important conclusion. How many RFs are observed typically in a single cell when they perform? How many RF-associated G4s are observed per cell. Just from looking at the images, I would imagine this would be a very small number, in single digits if not lower. Figure 1d is not very helpful because it expresses the G4 replication density per area, and does not show comparison between randomly assigned G4 replisome density and properly assigned.

2. Along the same line of discussion, it will be very helpful to provide a back of the envelope estimation of how many RFs are expected to be observed in a single imaging plane and what fraction of them are expected to contain G4 containing sequences. It is also very important to show that the fraction of G4 replisome among all replisomes observed increases with FANCI deletion, which is currently lacking in the paper. They only show FANCI's roles when G4 is chemically stabilized but not under native conditions.

3. In fact, lack of data showing that FANCI resolves G4 DNA under native condition is concerning. If FANCI is required for G4 resolution, when it is deleted, a higher fraction of replisomes should show G4 signal. Why do they not report these quantities when the measurements are performed without PDS?

4. In their smFRET studies, they were able to see that even 100 pM FANCI can resolve a stable G4 structure and load RPA onto it, which is very impressive. I wonder how the observations here are different from a related study from the Spies lab that the authors briefly mentioned here.

5. Figure 1g is used as evidence for G4 formation between CMG helicase and nascent DNA. However, because these spots are identified to be close to each other, Figure 1g alone is not convincing. They would need to plot the same but with EdU and G4 to align along the horizontal axis and show that MCM does not appear between the EdU and G4. They would also need to do the same but with MCM and G4 aligned along the horizontal axis and show that Edu does not appear between them.

6. Lines 177-179. They performed three color imaging of RPA/MCM/PCNA. How did they identify S phase cells if EdU imaging was performed for these experiments?

7. Line 190. "we plotted the overlaid TC triplets using MCM as the center of alignment". This choice seems arbitrary to me. Perhaps the authors can explain why this way of plotting was necessary.

8. Line 223. "In contrast, the brief 224 PDS treatment did not yield a noticeable change in nuclear G4 signal in siCTRL cells". This would suggest that FANCI would resolve PDS-stabilized G4s very well. But I would imagine this would be inconsistent with prior studies where higher G4 amount was observed with PDS.

9. Figure 3a-c. siFANCI did not increase G4 signal. Why?

10. Line 229. "These observations..." It is not clear where the data supporting the statement is presented. In a previously published paper?

11. Line 233. "Combined, these data demonstrate that the resolution of RF-coupled G4 formation and accumulation requires the helicase activity of FANCI." They showed this only under PDS treatment. They need to show this in the absence of PDS.

12. Line 242. "the level of EdU... remain unchanged". It is unclear unchanged by what. Unchanged by PDS treatment?

13. Figure 3h and i. Why is EdU not changed by siFANCI for G4 replisomes?

14. Line 245. "Taken together, our results demonstrate that RF-coupled G4 formation presents a spontaneous obstacle to replication fork progression and such events are further exacerbated in the case of FANCI deficiency." This can not be claimed to be spontaneous obstacle if PDS treatment is needed to see the effect of FANCI deficiency.

15. Figure 3f. Statistical significance in difference between siCTRL and siFANCI is questionable.

16. Line 272. "We found that PDS treatment led to a substantial enrichment in RPA binding at All-Replisomes in siCTRL cells, while siFANCI cells showed no significant change". If this is a property of native G4s, they should be able to show that RPA binding should decrease with siFANCI even in the absence of PDS treatment. As far as I can tell they do not show data on this. Without such data, the role of FANCI in loading RPA on native G4 is still unknown.

17. PDS treatment time varies between different experiments. Why? Do the conclusions depend on the treatment time?

18. PDS treatment was not used in the smFRET experiments whereas all FANCI dependencies in the cell were obtained with PDS treatment. So it is not yet certain that these two sets of experiments can be combined to reach a conclusion.

19. Line 377. "We therefore probed the DNA damage marker γ H2AX in U2OS cells via conventional

epifluorescence microscopy and noted a clear enhancement in γ H2AX foci in siFANCI cells compared to siCTRL cells following G4 stabilization." They would need to show this in cells that are not treated with PDS if they want to claim this is true for native G4 DNA.

20. Line 382. "a substantial accumulation of γ H2AX at G4-replisomes in PDS-treated siFANCI cells" It is not clear what is meant by substantial. G4 replisomes are very few in number to begin with. How many γ H2AX foci per cell?

21. Figure 6b. Similar plots should be shown for all replisomes. Also please explain how the data are normalized.

22. Figure 6d shows a model but there is no mention of PDS which is strange because a large number of conclusions in the paper are based on PDS-treated cells. Where does PDS play a role?

23. Line 463. "Our data revealed an increase in G4 formation frequency at replication forks upon FANCI depletion or inactivation" This is true only of PDS treated cells.

24. Is the fraction of G4 replisomes of 1.3% reasonable? What fraction of genome is being replicated at any give moment within S phase and among those what fraction would contain G4 forming sequences?

25. Line 494. "Accordingly, loss of FANCI results in accumulation of RPA-resistant G4s at forks" They don't show this.

26. Line 623. "second phase". Perhaps they meant second term? Same for first phase in the next line.

27. Line 683. "Otsu threshold" A reference may be needed here.

Reviewer #3 (Remarks to the Author):

In their manuscript, Lee et al use multi color SMLM and quantitative computational analysis to visualize the formation of G-quadruplex (G4) structures at replication forks during DNA replication. They show that G4 structures form at a subset of replisomes and hinder DNA synthesis and RPA binding. By knocking down the FANCI helicase, they show that RPA mediated FANCI binding is crucial for resolving G4 structures. In cells lacking FANCI, G4s accumulate leading to stalled replication and DNA damage. Finally using in vitro single molecule FRET assays they show that FANCI is essential for destabilizing folded G4 structures, supporting the cellular SMLM data.

Since my expertise on the biological problem studied here is very limited, I will mainly comment on the methodological approach. The authors use an interesting approach that they have developed to pull very quantitative information from their SMLM data essentially by taking advantage of spatial correlations among all the localized positions. I find the approach very exciting and commend the authors for being able to pull out such detailed information from essentially noisy data.

My one main concern is the labeling efficiency and the balance in the SMLM image among the various colors. It is well appreciated that various fluorophores don't all perform equally well in SMLM and the quality of the super-resolution image will be impacted by the fluorophore photoswitching properties. In addition, the various antibodies used may have different labeling efficiencies. Hence, I am wondering how much information may be missed because in each image only a fraction of each protein will be imaged and when you are analyzing three coupled proteins the problem is combinatorial. So a very small fraction of the spots will have all three proteins present merely because of missed localizations. It would be nice if the authors can comment on this point in the manuscript or carry out a control experiment where they label the same protein in three different colors to alleviate any concern that missed proteins will not impact the robustness of the results.

Point-by-Point Response to the Reviewers' Comments for NCOMMS-20-28452.

Single-Molecule Imaging Reveals Replication Fork Coupled Formation of G-quadruplex Structures Hinders Local Replication Stress Signaling

Wei Ting C. Lee, Yandong Yin, Michael J. Morten, Peter Tonzi, Diana C. Odermatt, Pam Pam Gwo, Mauro Modesti, Sharon B. Cantor, Kerstin Gari, Tony T. Huang, Eli Rothenberg*.

For clarity, the reviewers' comments are in **black** and our responses are marked in **blue**.

Reviewer #1 (Remarks to the Author):

This study is a technical tour-de-force providing the first direct proof for the presence of G4-DNA in the replication fork. This major advance has become possible by combining a multi-color single-molecule localization microscopy (SMLM) with unbiased automated pattern recognition protocol employing the Triple-Correlation (TC) function. In brief, nascent DNA in a human cell culture was pulse-labeled by EdU and co-stained with validated antibodies against the MCM-helicase, PCNA, RPA and G4-DNA. The S-phase nuclei were then selected and analyzed by the aforementioned approach. As the result, the authors came to the following conclusions:

- (1) Roughly 2% of all replisomes in human cells contained G4-DNA. The fraction of G4-containing replisomes was further increased 1.5-fold when the replicative helicase and DNA polymerases were partially uncoupled by treatment with mild aphidicolin concentrations.
- (2) Most importantly, the G4-DNA was located exactly between PCNA/nascent DNA and MCM signals. While this was suspected before, this is the first direct visualization of the G4-DNA location within the replisome. This observation alone warrants the publication in Nat Com.
- (3) Another very important observation is that G4-containing replisomes are deprived of RPA even under aphidicolin treatment, as opposed to the regular replisomes. This directly confirms that RPA alone does not efficiently bind to G-rich DNA stretches in ssDNA allowing them to fold into the G4-conformation.
- (4) The EdU incorporation within G4-containing replisomes appeared to be 20% lower than that in regular replisomes, supporting the long-suspected notion that once formed, G4-DNA inhibits further DNA synthesis.
- (5) Finally, the authors' siRNA experiments has demonstrated that depletion of FANCD1 DNA-helicase results in the enrichment of G4-containing replisomes. Their subsequent in vitro analysis by smFRET showed that FANCD1 unwinds G4-DNA starting from the loops, which in combination with RPA effectively disentangles these structures.

Overall, this is an outstanding study that covers all the bases. Nothing to add or subtract.

We thank the reviewer for their enthusiasm and support.

Reviewer #2 (Remarks to the Author):

We thank the reviewer for their comments and suggestions, and have revised the manuscript to address all the concerns that were raised, as detailed below. The revised manuscript now includes amended text and figures, additional data, analyses and controls, and expanded Supplementary Notes.

To better organize our reply and to avoid redundancy in our response we have collated some of the reviewer's remarks such that our replies are based on three key issues raised:

A. Fraction of forks with G4 signals (#1, 2A, 24);

B. Comparison between native siCTRL vs siFANCI cells (#2B, 3, 9);

C. Chemically stabilized vs native G4s (#2C, 11, 14, 16, 18, 19, 22, 23, 25).

We had grouped those comments together and provided an overall response for each group, followed by specific response for individual questions. For clarity, we have also separated the reviewer's comment #2 into three parts (2A, 2B, and 2C) based on their respective relevance to the three key issues as categorized above:

2A. Along the same line of discussion, it will be very helpful to provide a back of the envelope estimation of how many RFs are expected to be observed in a single imaging plane and what fraction of them are expected to contain G4 containing sequences.

2B. It is also very important to show that the fraction of G4 replisome among all replisomes observed increases with FANCI deletion, which is currently lacking in the paper.

2C. They only show FANCI's roles when G4 is chemically stabilized but not under native conditions.

A. Fraction of forks with G4 signals

- 1. Line 62. direct evidence as to the formation of RF coupled G4s: This is a key point of the manuscript, and needs to be firmly established. Only 2.2% of observed RFs show G4 signal. This is probably not surprising because one would assume only a small fraction of RFs have an unresolved G4, either because there is no G4 forming sequencing there or because it has already been resolved. However, such a small fraction raises some concerns about the robustness of this important conclusion. How many RFs are observed typically in a single cell when they perform? How many RF-associated G4s are observed per cell. Just from looking at the images, I would imagine this would be a very small number, in single digits if not lower. Figure 1d is not very helpful because it expresses the G4 replication density per area, and does not show comparison between randomly assigned G4 replisome density and properly assigned.
- 2A. Along the same line of discussion, it will be very helpful to provide a back of the envelope estimation of how many RFs are expected to be observed in a single imaging plane and what fraction of them are expected to contain G4 containing sequences.
- 24. Is the fraction of G4 replisomes of 1.3% reasonable? What fraction of genome is being replicated at any give moment within S phase and among those what fraction would contain G4 forming sequences?

To address this point we have revised the manuscript, which now includes extended descriptions of how our observations relate to other studies and estimates thereof, by providing an estimation of the percent-range of G4-positive RFs that are expected to be observed in a single cell, derived from computational predictions of genomic G4 motifs, genomic data, and live-cell imaging studies (Supplementary Note 2). We note that the percentage of G4-positive RFs we observed are well within these estimates. Specifically, our estimates are based on the following:

- a. **Estimates based on genomic data:** In terms of the prevalence of G4-containing sequence, genomic analyses provided an estimate of approximately 700,000 G4 motifs with elaborate structural considerations (Chambers et al., 2015; Puig Lombardi et al., 2019). However,

given the sequence variability and thermodynamic stability of G4s, as well as the complexity of cellular environment, the majority of these motifs are unlikely to form G4 structures in cell, and when formed they can be readily destabilized by DNA binding proteins such as RPA. Of the overall population of G4 motifs, it was found that there is an increase representation for G4 motifs with short-loops. Specifically, in the human genome, single-nucleotide loop motifs (G4-L1) are statistically overrepresented and make up ~5% of all G4 motifs (Puig Lombardi et al., 2019). Of the total G4-L1 motifs, the thermal stabilities (experimental $T_{1/2}$, derived by their melting curves) of G4-L1A (A-A-A loops) and mixed-base loops are not likely to pose an issue on genomic integrity as they can be destabilized by RPA, whereas G4-L1T, G4-L1G and G4-L1C (T-T-T, G-G-G, and C-C-C loops) are more stable ($60\text{ }^{\circ}\text{C} < T_{1/2} < 65\text{ }^{\circ}\text{C}$) and therefore can contribute to genomic instability, as shown by the Nicolas lab (Piazza et al., 2015). While the total number of G4-L1 motifs in the human genome is estimated to be close to 39,000, the more thermodynamically stable G4-L1 motifs (G4-L1T, G4-L1G and G4-L1C) with higher chance of stably forming into a G4 in cell are only ~9% of that population, which is approximately ~3,500 motifs (Puig Lombardi et al., 2019). Considering the genome size to be ~ 6 billion base pairs, with the average size of a replicon to be 100-120kb (Mechali, 2010), each RF is about 50-60kb and ~100,000 forks in total are needed to replicate the entire genome in one cell. Based on this, we estimate that a total of **3.5%** (3,500 G4 motifs/100,000 forks) of forks could ‘encounter’ a stable G4-L1 motifs throughout the course of the entire S phase. The actual fraction would vary from early to late S-phase with changes in replication pace, distribution and whether the stable G4-L1 motifs actually fold into G4 structures. Thus, the percentage of observed forks with G4 structures is expected to be in the range of **2-3%**.

- b. **Estimates based on G4-ChIP-seq measurements:** A recent study using G4-ChIP-seq had detected ~1,000 to ~10,000 G4 in different human cell-lines (Hansel-Hertsch et al., 2016). Using these values, we estimate the probability of observing a G4-RF to be **1-10%**.
 - c. **Estimates based on live cell studies:** In a very recent paper utilizing elegant detection of G4s in live cells, the authors have had estimated a total number of G4s in a single U2OS cell of ~3000 (Di Antonio et al., 2020). Using this value, the fraction of G4-RF would be **~3%**.
- For calculating the fraction of G4-positive forks from our SMLM images, we used the DBSCAN clustering and Nearest Neighboring Distance (NND) approach, which is a threshold-based method for calculating colocalizations between two molecules-of-interest, and have been widely applied in image analysis of localization data (Coltharp et al., 2014; Nicovich et al., 2017). While our calculation (~2.24%) is consistent with the estimates provided above, we would like to emphasize that our calculations are only mean to provide an *approximation* rather than an absolute number.
 - In our study we also utilized the probabilistic Triple-Correlation (TC) analysis approach, in addition to DBSCAN and NND techniques, for determining the presence and relative frequencies of RF-coupled G4s. The TC approach was developed by our lab (Yin et al., 2019; Yin and Rothenberg, 2016) and is uniquely suitable for defining and quantifying distinct molecular patterns within complex multiplexed SR images such as the nucleus, in an unbiased manner. To the best of our knowledge, this is the only approach for unbiased statistical data-mining of complex molecular configurations in multi-color SMLM images. Specifically, in this manuscript, we utilized this approach to determine if there are any spatial correlations among G4 and replisome complexes (MCM and EdU). We further elaborate that the specific strengths of TC approach include the following:
 - 1) TC preforms unbiased data-mining since it samples the significance of a triplet pattern from the frequency domain without any user-defined clustering/colocalization thresholding, and no pre-conceived patterns are needed;

2) TC computes the spatial correlation among G4-MCM-EdU against the hypothesis that G4, MCM, EdU are randomly distributed. Figure 1g and h reveals that in intact cells, the labeled G4s show significant correlation to the MCM-EdU pair rather than randomly distribute around MCM-EdU, meaning that G4 are highly probabilistically found at MCM-EdU forks regardless how low the G4-replisome fraction might be.

- To further elucidate and characterize the observed association and specific patterns, with the hypothesis that G4 is folded at ssDNA, we treated the cells with APH and observed a further increase in such 3-component association, as shown in figure 1h. We conclude that G4 can form within replisomes as ssDNA is exposed upon MCM helicase unwinding. This provides, to the best of our knowledge, the first direct quantitative visualization of G4 localizing within the replisome, addressing the previously unanswered question of how G4 can interact with and affect the replisome.
- We noticed that the way we introduce the DBSCAN/NND data may be off-focus. We therefore have modified the description of this method and result in the main text in line ~100/131. In addition, to further elaborate on our estimates, we included the above discussion in Supplementary Note 2.

B. Comparison between native siCTRL vs siFANCJ cells

- 2B It is also very important to show that the fraction of G4 replisome among all replisomes observed increases with FANCJ deletion, which is currently lacking in the paper. They only show FANCJ's roles when G4 is chemically stabilized but not under native conditions.
- 3. In fact, lack of data showing that FANCJ resolves G4 DNA under native condition is concerning. If FANCJ is required for G4 resolution, when it is deleted, a higher fraction of replisomes should show G4 signal. Why do they not report these quantities when the measurements are performed without PDS?
- 9. Figure 3a-c. siFANCJ did not increase G4 signal. Why?

To address these points, we have revised the manuscript which now provides the following clarifications and analyses:

- We emphasize that the depletion of FANCJ in U2OS cells did result in an enrichment in nuclear G4s, which is shown in Figure S2b. We note that this enrichment was further elevated upon PDS exposure, as shown in Figure S2b (Line 214).
- To determine whether the observed increase in G4 structures stems from the stabilization of RF-coupled G4s, we used both DBSCAN/NND and TC as independent approaches for quantifying the relative G4-Replisome frequencies in siCTRL and siFANCJ cells. The fractions of G4-positive-replisome in siCTRL vs siFANCJ cells, calculated by DBSCAN/NND, is shown in Figure S2c. We noted that although the nuclear density of G4 shown an increase in NT siFANCJ cells compared to NT siCTRL cells (Figure S2b), the fraction of PCNA associated with G4 is lower in NT siFANCJ cells compared to NT siCTRL cells (Figure S2c). We therefore quantified the level of active replication via AC analysis of EdU (Figure S2e) and of chromatin bound PCNA (Figure S2f), each serve as a distinct marker of active replications forks. These results revealed an increase in the level of active replication in siFANCJ cells as compared to siCTRL cells. This increase is anticipated due to the previously reported role of FANCJ in regulating origin firing and replication initiation (Greenberg et al., 2006; Raghunandan et al., 2015), where FANCJ-depletion lead to increase in origin firing and active forks, shown as elevated EdU and PCNA levels. The overall increase in active forks would therefore lead to a decrease in the relative G4-replication fraction in siFANCJ cells. The data and above explanations are now included in Figure S2.
- Since the difference in the level of active replication fork in siCTRL vs siFANCJ cells would contribute to the relative change in the frequency of replisomes that are G4-positive, we decided to normalize the TC data to the respective NT condition/population in siCTRL cells and in siFANCJ

cells, which enable us to resolve the specific effects resulting from G4s with high thermostability. Specific points raised regarding the effect PDS in this context are addressed below in C.

C. Chemically stabilized vs native G4s

- 2C. They only show FANCI's roles when G4 is chemically stabilized but not under native conditions.
- 11. Line 233. "Combined, these data demonstrate that the resolution of RF-coupled G4 formation and accumulation requires the helicase activity of FANCI." They showed this only under PDS treatment. They need to show this in the absence of PDS.
- 14. Line 245. "Taken together, our results demonstrate that RF-coupled G4 formation presents a spontaneous obstacle to replication fork progression and such events are further exacerbated in the case of FANCI deficiency." This can not be claimed to be spontaneous obstacle if PDS treatment is needed to see the effect of FANCI deficiency.
- 16. Line 272. "We found that PDS treatment led to a substantial enrichment in RPA binding at All-Replisomes in siCTRL cells, while siFANCI cells showed no significant change". If this is a property of native G4s, they should be able to show that RPA binding should decrease with siFANCI even in the absence of PDS treatment. As far as I can tell they do not show data on this. Without such data, the role of FANCI in loading RPA on native G4 is still unknown.
- 18. PDS treatment was not used in the smFRET experiments whereas all FANCI dependencies in the cell were obtained with PDS treatment. So it is not yet certain that these two sets of experiments can be combined to reach a conclusion.
- 19. Line 377. "We therefore probed the DNA damage marker γ H2AX in U2OS cells via conventional epifluorescence microscopy and noted a clear enhancement in γ H2AX foci in siFANCI cells compared to siCTRL cells following G4 stabilization." They would need to show this in cells that are not treated with PDS if they want to claim this is true for native G4 DNA.
- 22. Figure 6d shows a model but there is no mention of PDS which is strange because a large number of conclusions in the paper are based on PDS-treated cells. Where does PDS play a role?
- 23. Line 463. "Our data revealed an increase in G4 formation frequency at replication forks upon FANCI depletion or inactivation" This is true only of PDS treated cells.
- 25. Line 494. "Accordingly, loss of FANCI results in accumulation of RPA-resistant G4s at forks" They don't show this.

To address the specific points raised by the reviewer regarding the effect of PDS-stabilized vs native G4s, we have revised the manuscript for further clarifications. Below, we provide additional scientific background and highlighted our strategy, rationale and description of our SMLM experimental approach to probe the effects of G4s with high thermostability on replication by enrichment via PDS, as well as the use of specific G4 structures with known thermal stability for our *in-vitro* smFRET studies:

Background:

- Several *in vitro* studies on the characteristics of G4 structures have shown that the stability of G4 structures is affected by the length and sequence of the loop, wherein shorter loops, especially single-base T-T-T loops, result in more stable G4 structures (Hazel et al., 2004; Rachwal et al., 2007).
- Bioinformatics studies that analyzed the occurrences of G4 motifs across species (Puig Lombardi et al., 2019) found an increased bias for G4 motifs with short-loops. Specifically, in the human genome, single-nucleotide loop motifs (G4-L1) make up 5% of the total G4 motifs. As discussed above in (A), of the total G4-L1 motifs, the thermal stability of G4-L1A and mixed-base loops is less than 50°C, and therefore will not pose any issue on genomic integrity

and are likely to be destabilized by RPA. In contrast, G4-L1T, G4-L1G and G4-L1C are more stable ($60\text{ }^{\circ}\text{C} < T_{1/2} < 65\text{ }^{\circ}\text{C}$) and therefore can contribute to genomic instability as shown by the Nicolas lab (Piazza et al., 2015). The over-representation of short-loop G4s in the human genome (as compared to other possible iterations) indicates that short-looped G4s have been subjected to selective pressure that favors their emergence and maintenance in genomes, possibly due to their potential biological functions (transcription, replication, epigenetic regulation, etc). This, however, also suggests that the more stable G4-L1 structures, G4-L1T, G4-L1G and G4-L1C, would require regulatory mechanisms in order to sustain a balance between their active roles and their deleterious effects to the genome.

- A comprehensive study by the Balci lab had used smFRET assays and other biophysical approaches to systematically characterize the binding and unfolding ability of RPA on diverse G4 structures, showing that stable short loop G4 structures (such as G4-L1T) are remarkably resistant to RPA binding even at physiological concentration of RPA ($\sim 1\text{ }\mu\text{M}$), whereas other G4 structures with lower thermal stability and/or longer loop length can be readily unfolded by RPA at nM range (Ray et al., 2013).
- Based on these studies, and given the cellular concentration of RPA, most transient G4 structures that form during replication can be efficiently unfolded by RPA alone and therefore will not present any issue to replication fork progression. However, a small population of G4s (about 9% of the total G4-L1) that are more thermodynamically stable, such as G4-L1T, G4-L1G and G4-L1C, will require additional factors to assist in their unfolding as they cannot be destabilized by RPA alone. We therefore focused on the contribution of one such factor, the FANCD1 helicase.

Our *in vitro* smFRET measurements (Figure 5) were devised to address the following question: *Can the binding of RPA onto a highly stable G4 structure be assisted by FANCD1?*

- In these assays, we directly probe the loading of RPA onto stable G4s by designing a single-looped G4 structure with high thermal stability (G4-L1T, with T-T-T loops) and compared it to a G4 structure with a longer loop and lower thermal stability (G4-L3 with TTA-TTA-TTA loop) (Figure 5 and Figure S4). We emphasize that in these experiments we directly measured the unfolding of G4 structures (mediated by RPA and/or FANCD1) as a function of their known thermal stabilities. We emphasize that the binding of G4 ligand to G4 would simply increase the melting temperature of the structure by $\sim 10^{\circ}\text{C}$, thus elevating the thermal stability of the structure. In other words, in the presence of a G4 ligand the stability of a short-loop G4-L1A would increase and be comparable to the baseline thermal stability of a G4-L1T structure (De Cian et al., 2005; Marchand et al., 2018; Puig Lombardi et al., 2019). Given that the direct nature of our smFRET measurements relates to the known melting temperature of the G4 structures, adding any G4 stabilizing ligand will be unwarranted as it would have been redundant with the direct measurements we performed.
- It is also important to note that several studies had investigated how PDS and other G4 stabilizing ligands affect the G4 unwinding activity of G4 helicases *in vitro* (Chen et al., 2015a; Maleki et al., 2019), showing that the increase in G4 stability due to G4 stabilizing ligands is equivalent to that of more stable G4s without the ligand, as they do not affect nor abolish the activity of G4 helicases nor ATP hydrolysis, but rather reduce the ability of helicases to efficiently unwind the G4 structure and stabilize the unfolded conformation, which is akin to the more thermodynamically stable native G4s, and in agreement with our observations from the smFRET study.

Cellular measurements of fork-coupled G4s in the presence of PDS:

- As mentioned above, the formation of thermodynamically stable G4s at forks has an overall low occurrence, with only $\sim 2\%$ of forks are found to be associated with G4s in our measurements, which is in agreement with the above estimates and studies by other

labs (Di Antonio et al., 2020; Hansel-Hertsch et al., 2016; Puig Lombardi et al., 2019). The low probability of observing stable native G4 structures at replication forks in cells (as also indicated by the reviewer) explains why the differences in RPA binding in NT-siCTRL vs NT-siFANCI cells (#16 of reviewer's comment), or γ H2AX foci in NT-siFANCI cells (#19 of reviewer's comment) are not readily detected using conventional fluorescence microscopy. In our study, we aim to probe how stable native G4s that are known to counteract RPA *in vitro* affect replication forks in cell. Given the generally low incidence of these structures *in vivo* we briefly treated cells with the G4-stabilizing ligand PDS. We note that the addition of PDS only increases the melting temperature (thermostability) (Rodriguez et al., 2008) of already folded native G4s, thereby slightly enrich the frequency of stable G4s at forks, providing improved probing of their presence and effects on replication forks. By increasing the incidence of stable G4s at forks, we were able to refine our observations with respect to the role of FANCI in assisting the loading of RPA onto these structures, as presented in our model illustrated in Figure 6.

To clarify our observations and rationale with respect to the effect of stable native G4s at forks in the presence of PDS, we have made modifications in the method (line 506), results and discussion sections to note 1) that our study focuses on the effects and regulations of RF-coupled G4s that have high thermostability and RPA resistance; and 2) the rationale behind the usage of PDS. In addition, in response to reviewer's specific comments, we have made the following edits:

1. For #11, We have modified the sentence to: "Combined, these data demonstrate that the resolution of stable G4s that form at replication forks requires the helicase activity of FANCI." (Line 225)
2. For #14, We have deleted the word "spontaneous". (Line 236)
3. For #22, we modified the model to emphasize the collaboration between RPA and FANCI in regulating stable G4s, which are RPA resistant, as compared to other G4s that are readily resolved by RPA alone. (Figure 6)

4. In their smFRET studies, they were able to see that even 100 pM FANCI can resolve a stable G4 structure and load RPA onto it, which is very impressive. I wonder how the observations here are different from a related study from the Spies lab that the authors briefly mentioned here.

- In the smFRET study reported by the Spies lab (Wu and Spies, 2016), when adding 100pM FANCI (+ATP) to G4, they did not show persistent unfolding of G4 by FANCI. Instead, they observed repetitive unfolding and refolding of their G4 structures. Using the rigorous hidden Markov modeling approach, they identified several intermediate FRET states between the folded and unfolded states, indicated that FANCI-mediated G4-unfolding proceeded through different intermediates that could represent distinctive partially-folded G4 structures. They proposed that the partial unfolding-refolding cycles is needed to keep the FANCI protein nearby G4 until the translesion polymerase REV1 is recruited for DNA synthesis (Eddy et al., 2014; Sarkies et al., 2012).
- In our experiments, we used same concentration of FANCI as used in the Speis' study. In agreement with their observation, we did not observe any persistent unfolding when FANCI was added without RPA (Figure 5b [-RPA +FANCI]). However, it is important to distinguish that the design of the G4 substrates in our study differs from the those used in Spies' study. The G4 substrates used in their study contained a ssDNA stretch at the 5' side of the G4 structure, which served as a "loading window" for FANCI, and a duplex DNA at 3' of the G4 structures. The substrate we used contained ssDNA at both sides of the G4 structure to reflect a G4 that form along ssDNA. Since FANCI helicase has a 5' to 3' polarity, having a ssDNA at both sides of G4 may therefore provide an additional binding coordination for FANCI. This, however, did not improve the binding and subsequent G4 unwinding by RPA in the absence of FANCI.

- In our experiments, persistent unfolding of the G4 structure was only observed when we added RPA along with 100pM FANCI +ATP. Our observations seem to be in line with the model proposed by the Spies lab, where stable unfolding and/or processing of G4s would require other proteins in addition to the FANCI helicase. Complementary processing of G4 structure in cells could involve FANCI and RPA which collaborate to unwind stable G4 structures, while REV1 (and /or other TLS or replicative polymerases) are recruited for proper DNA synthesis.

5. Figure 1g is used as evidence for G4 formation between CMG helicase and nascent DNA. However, because these spots are identified to be close to each other, Figure 1g alone is not convincing. They would need to plot the same but with EdU and G4 to align along the horizontal axis and show that MCM does not appear between the EdU and G4. They would also need to do the same but with MCM and G4 aligned along the horizontal axis and show that Edu does not appear between them.

- We have elaborated on the plotting of TC triplets in Supplementary Note 3, which is also detailed in our previous publication (Yin et al., 2019). We emphasize that the obtained triplet pattern from a single nucleus represents an averaged representation of the distance correlation among all the non-random, triple-component molecular complexes identified within the given correlation range. We note that when the scale of the pattern is comparable to or smaller than the combined spatial resolution (~40 nm), the pattern tends to present as an equilateral triangle (examples shown in Figure 1b and e, or Supplementary Note 3 Figure 1c), due to the disparity between the center-to-center distance and the average distance among the three co-localized clusters that can overlap with each other (see Supplementary Figure 5 in (Yin et al., 2019) for more detailed explanation). However, it is important to note that the compact triplet pattern do not affect the quantifications of the specific components within each pattern (Supplementary Note 3 Figure 2 and 3; Yin et al., 2019).
- To provide experimental validation for G4 formation between the CMG helicase and nascent DNA, we designed experiments in which we directly induce the uncoupling of CMG from the polymerase by treating cell with APH, which results in more ssDNA within the replisomes that was expected to form more G4s. In agreement with our hypothesis, our analyses were able to detect the specific increase of G4-replisomes (Figure 1h and i), indicating that DNA G4s can form at forks during replication progression as the ssDNA is exposed between CMG and nascent, illustrated in Figure 1f. It is also important to note that G4-Replisomes induced by APH treatment exhibit the same distinct behavior as the G4-Replisomes observed in untreated cells (Figure 2).
- Finally, we emphasize that throughout the study, the magnitudes of any TC triplets derived from individual nuclei are provided via TC quantifications, whereas the TC triplet plots (overlays) serve as a visual representation. It is important to note that variations in distance configurations and pattern alignment do not affect the quantifications of the specific components within each pattern.
- We have elaborated on this point on Supplementary Note 3, while modified the main text of the manuscript for clarification.

6. Lines 177-179. They performed three color imaging of RPA/MCM/PCNA. How did they identify S phase cells if EdU imaging was performed for these experiments?

- It is long-established that PCNA serves as a reliable S-phase marker, and has been frequently used to discriminate different cell cycle phases in countless cellular studies (Dimitrova et al., 1999; Kisielewska et al., 2005; Schonenberger et al., 2015). Therefore, in our studies, we used PCNA and EdU as independent S-phase markers. We have made modification in line 212 for clarification.

7. Line 190. "we plotted the overlaid TC triplets using MCM as the center of alignment". This choice seems arbitrary to me. Perhaps the authors can explain why this way of plotting was necessary.

- We elaborate on the plotting of TC triplets in response #5, Supplementary Note 3, and in our previous studies (Yin et al., 2019). We emphasize that throughout the study, the magnitudes of any TC triplets derived from individual nuclei are provided via TC quantifications, whereas the TC triplet plots (overlays) serve as a visual representation. Specifically, for all plots related to the measurement of EdU incorporation we present the TC triplets this way to help visualize the relative abundance of EdU at forks under different conditions (Figures 2e, 3d, 3g). It is important to note that variations in distance configurations and pattern alignment do not affect the quantifications of the specific components within each pattern.

8. Line 223. "In contrast, the brief 224 PDS treatment did not yield a noticeable change in nuclear G4 signal in siCTRL cells". This would suggest that FANCI would resolve PDS-stabilized G4s very well. But I would imagine this would be inconsistent with prior studies where higher G4 amount was observed with PDS.

- In our studies we focused on the formation of G4s that are coupled to replication fork. We therefore only imaged and analyzed S-phase cells (EdU and/or PCNA positive) with or without 1hr of PDS treatment, with the rationale being that the brief time window of drug treatment would allow for stabilizing some folded G4 structures that are formed during replication fork progression.
- We emphasize that our observations with respect to the effect of 1-hr PDS treatment in siCTRL cells are in fact consistent with data from a recent paper by Giovanni Capranico and colleagues (De Magis et al., 2019). In their study, they performed a refined temporal analysis of the cellular response to treatment with G4 ligands including PDS, revealing a rapid increase in the number of nuclear G4s at 2-10 min following treatment, which then dropped to baseline levels at 30-60 min; (see Figure 1 of (De Magis et al., 2019)), in agreement with our observations. Based on these data, the authors concluded that the immediate increase in G4s induced by PDS at 2-10 min is rapidly followed by cellular response that facilitate the resolution of G4 structure, likely by G4 specific helicases, enabling cellular G4s to return to baseline levels as early as 30-60 min following treatment. In accordance with their model, our study suggests that RPA and the FANCI helicase is involved in promoting timely G4-resolution.
- It is important to distinguish that while other studies have mainly examined the effect of chronic G4 stabilization by ligands treatment for 24 hrs or more (Balasubramanian et al., 2011; Hansel-Hertsch et al., 2017), the timeline and cellular response to G4s ligands is still a matter of ongoing studies. Notably, longer treatments can induce G4 related events that occur outside S-phase and are independent of replication.

10. Line 229. "These observations???" It is not clear where the data supporting the statement is presented. In a previously published paper?

- To address this point, we have modified the text and included appropriate citation on line 220

12. Line 242. "the level of EdU??? remain unchanged". It is unclear unchanged by what. Unchanged by PDS treatment?

- We have modified the text to clarify this statement on line 233.

13. Figure 3h and i. Why is EdU not changed by siFANCI for G4 replisomes?

- A key advantage of TC analysis is its ability to measure the unique properties of *G4-replisome* (EdU/MCM/G4) as opposed to *All-replisome* (EdU/MCM/PCNA). In Figure 3h and i, we quantified the local density of EdU associated with replisome that are G4-positive. We consistently

observe low levels of EdU incorporation at G4-replisomes, which indicates that G4 formation at forks is sufficient to impede DNA synthesis locally at those sites (Figure 3h and i, as well as Figure 2f for comparison between All- vs. G4-replisomes). Note that this phenotype of low EdU incorporation observed for G4-replisomes is irrespective of the relative abundance of G4-replisomes in each condition. However, increased G4-Replisomes in PDS-treated siFANCJ cells (Figure 3a-c) means more forks exhibit reduced EdU incorporation. As a result, we observed a decrease in the level of EdU incorporation calculated for *All*-replisome (Figure 3d-f).

15. Figure 3f. Statistical significance in difference between siCTRL and siFANCJ is questionable.

- We note that Figure 3d-f showed the local densities of EdU at All-Replisomes in NT vs PDS-treated siCTRL or siFANCJ cells. We clarify that the data presented in Figure 3f was directly derived from data from Figure 3e. In Figure 3e, we performed unpaired two-sample t-test between NT vs PDS conditions in siCTRL or siFANCJ cells respectively, with sample sizes of $N = 53, 47, 47, 46$. Our results indicated that the change of EdU local densities upon PDS exposure is non-significant ($p = 0.21$) in siCTRL cells, but significant ($p = 0.01$) in siFANCJ cells. For Figure 3f, we calculated the percent-change between the respective PDS-treated and NT conditions [(PDS-NT/NT)*100%] using the mean values obtained from Figure 3e. The errors in Figure 3f are propagated from those in Figure 3e.

17. PDS treatment time varies between different experiments. Why? Do the conclusions depend on the treatment time?

- To capture G4s that form during replication fork progression, we treated the cells with 20uM PDS for a short period of 1 hour and focused on S-phase cells for analyses (please see response for #8 for details). As shown in Figure 3a-c and S2b-c, we determined these conditions to be sufficient for inducing changes in the abundance of G4-Replisomes in siFANCJ cells that can be detected via our experimental approach.
- The experiments shown in Figure 4 (RPA recruitment) presents measurements that reflect the cellular response to formation of stable G4s at forks, which contrast the defective response in siFANCJ cells. When using the minimal treatment conditions above we did not observe a marked change in the levels of RPA in either siCTRL or siFANCJ cells, which is anticipated given the rapid cellular response in resolving stable G4s shortly after PDS treatment (see Figure 1 of (De Magis et al., 2019)). Based on this, it was projected that longer durations of PDS treatments was needed to reveal any possible changes. Since we sought to avoid chronic conditions, we extended the duration of PDS treatment and found that 4 hours of treatment are sufficient to induce an increase in RPA signal at All-Replisomes in siCTRL cells but not in siFANCJ cells (Figure 4a-c; Supplementary Figure 3a). For clarity, we also included the above discussion and the 1hr PDS data for RPA binding in Supplementary Figure S3c and d.

20. Line 382. "a substantial accumulation of γ H2AX at G4-replisomes in PDS-treated siFANCJ cells" It is not clear what is meant by substantial. G4 replisomes are very few in number to begin with. How many γ H2AX foci per cell?

- In these experiments we used the TC approach to quantify the *local* density of γ H2AX specifically at G4-replisome (i.e. conditional probability density of γ H2AX that is correlated to an G4-MCM pair), irrespective of the absolute abundance of G4-Replisome in each cell. This analysis revealed a significant increase in γ H2AX accumulation locally at G4-replisomes in PDS-exposed siFANCJ cells as compared to untreated cells. We emphasize that this analysis is threshold free, and does not utilize any DDR foci counting approaches.

- We have modified the text for clarity. We have also included data for All-Replisomes (See #21) to emphasize the difference between G4-Replisomes vs. All-Replisomes.

21. Figure 6b. Similar plots should be shown for all replisomes. Also please explain how the data are normalized.

- We have included our data showing γ H2AX at All-Replisomes in Supplementary Figure S5d and modified the text accordingly (line 374). We note that we did not observe any significant changes in γ H2AX density at All-Replisomes in both siCTRL and siFANCI cells upon PDS treatment, which indicates that the γ H2AX signals we observe at G4-Replisomes arises from few events that are specifically localized at G4-Replisomes (remark #20) and therefore would be averaged out over the entire population. The data was normalized, as detailed above in response #B, to the respective NT condition (NT-siCTRL or NT-siFANCI cells).

26. Line 623. "second phase". Perhaps they meant second term? Same for first phase in the next line.

- We have changed "phase" to "term". (Line 620)

27. Line 683. "Otsu threshold" A reference may be needed here.

- We have included a reference for the Otsu threshold. (Line 680)

Reviewer #3 (Remarks to the Author):

In their manuscript, Lee et al use multi color SMLM and quantitative computational analysis to visualize the formation of G-quadruplex (G4) structures at replication forks during DNA replication. They show that G4 structures form at a subset of replisomes and hinder DNA synthesis and RPA binding. By knocking down the FANCD1 helicase, they show that RPA mediated FANCD1 binding is crucial for resolving G4 structures. In cells lacking FANCD1, G4s accumulate leading to stalled replication and DNA damage. Finally using in vitro single molecule FRET assays they show that FANCD1 is essential for destabilizing folded G4 structures, supporting the cellular SMLM data.

Since my expertise on the biological problem studied here is very limited, I will mainly comment on the methodological approach. The authors use an interesting approach that they have developed to pull very quantitative information from their SMLM data essentially by taking advantage of spatial correlations among all the localized positions. I find the approach very exciting and commend the authors for being able to pull out such detailed information from essentially noisy data.

My one main concern is the labeling efficiency and the balance in the SMLM image among the various colors. It is well appreciated that various fluorophores don't all perform equally well in SMLM and the quality of the super-resolution image will be impacted by the fluorophore photoswitching properties. In addition, the various antibodies used may have different labeling efficiencies. Hence, I am wondering how much information may be missed because in each image only a fraction of each protein will be imaged and when you are analyzing three coupled proteins the problem is combinatorial. So a very small fraction of the spots will have all three proteins present merely because of missed localizations. It would be nice if the authors can comment on this point in the manuscript or carry out a control experiment where they label the same protein in three different colors to alleviate any concern that missed proteins will not impact the robustness of the results.

We thank the reviewer for their positive feedback. Indeed, the reviewer raises an important point that is highly relevant to the SMLM imaging community, where optimal labeling efficiency, photoswitching properties, and analysis approaches remain amongst the key challenges for multi-color STORM imaging. These issues have been considered throughout this study as well as in our previous studies (Whelan et al., 2020; Whelan et al., 2018). We therefore revised the manuscript to address and clarify this specific point as detailed below:

- Firstly, we would like to emphasize that the specific objective for utilizing multi-color SMLM approach in this study is to acquire a representative sampling for the nanoscale localization of specific molecular targets and complexes, rather than to resolve a refined molecular structure. Thus, for each target we optimized the sample labeling in order to enable significant detection with efficient single-molecule blinking. Since replisome complexes inside the nucleus are abundant, our experimental conditions were more than sufficient to provide a representative sampling of adequate triplets with significant associations, allowing us to elucidate the characteristics of our triplets-of-interest. We note that if insufficient triplets were labeled, the TC analysis would have not yielded any significant associations that are above the background of random associations.
- To further address the specific point raised by the reviewer, we included the control experiments suggested by the reviewer. We analyzed the triple-correlation of EdU+PCNA (both served as replication fork markers, **Supplementary Note 3 Fig. 4a**) or RPA (**Supplementary Note 3 Fig. 4b**) were triple-stained in different colors. We found their resulting triple-correlation magnitude, regardless of the color combination, to be significantly stronger than the correlation magnitude obtained from simulated images with random colocalizations (see method and Fig. S3 and Fig. S4 in Chen, Y. H. *et al.* (Chen et al., 2015b) for randomization procedure). These measurements indicate that the TC approach is capable of robust determination of distinct molecular

configurations despite potential under-sampling due to different fluorophore photoswitching properties, labeling efficiencies, or the crowded environment of our region of interest (the nucleus). It is important to note that the distinct configurations are well resolved notwithstanding the expected changes in the absolute TC magnitudes obtain for different color combinations. Finally, we note that in order to minimize errors and variabilities that may arise from different blinking and labeling biases we thoroughly test, validate and optimize the labeling conditions for each molecule-of-interest. Once these conditions are established, we maintain these conditions throughout our experiments by using the same antibody dilutions and fluorophore conjugations for the same targets to avoid any comparison of the same target between different antibodies and different colors (**Supplementary Table S2**).

Balasubramanian, S., Hurley, L.H., and Neidle, S. (2011). Targeting G-quadruplexes in gene promoters: a novel anticancer strategy? *Nat Rev Drug Discov* *10*, 261-275.

Chambers, V.S., Marsico, G., Boutell, J.M., Di Antonio, M., Smith, G.P., and Balasubramanian, S. (2015). High-throughput sequencing of DNA G-quadruplex structures in the human genome. *Nat Biotechnol* *33*, 877-881.

Chen, M.C., Murat, P., Abecassis, K., Ferre-D'Amare, A.R., and Balasubramanian, S. (2015a). Insights into the mechanism of a G-quadruplex-unwinding DEAH-box helicase. *Nucleic Acids Res* *43*, 2223-2231.

Chen, Y.H., Jones, M.J., Yin, Y., Crist, S.B., Colnaghi, L., Sims, R.J., 3rd, Rothenberg, E., Jallepalli, P.V., and Huang, T.T. (2015b). ATR-mediated phosphorylation of FANCI regulates dormant origin firing in response to replication stress. *Mol Cell* *58*, 323-338.

Coltharp, C., Yang, X., and Xiao, J. (2014). Quantitative analysis of single-molecule superresolution images. *Curr Opin Struct Biol* *28*, 112-121.

De Cian, A., Guittat, L., Shin-ya, K., Riou, J.F., and Mergny, J.L. (2005). Affinity and selectivity of G4 ligands measured by FRET. *Nucleic Acids Symp Ser (Oxf)*, 235-236.

De Magis, A., Manzo, S.G., Russo, M., Marinello, J., Morigi, R., Sordet, O., and Capranico, G. (2019). DNA damage and genome instability by G-quadruplex ligands are mediated by R loops in human cancer cells. *Proc Natl Acad Sci U S A* *116*, 816-825.

Di Antonio, M., Ponjavic, A., Radzevicius, A., Ranasinghe, R.T., Catalano, M., Zhang, X., Shen, J., Needham, L.M., Lee, S.F., Klenerman, D., *et al.* (2020). Single-molecule visualization of DNA G-quadruplex formation in live cells. *Nat Chem* *12*, 832-837.

Dimitrova, D.S., Todorov, I.T., Melendy, T., and Gilbert, D.M. (1999). Mcm2, but not RPA, is a component of the mammalian early G1-phase prereplication complex. *J Cell Biol* *146*, 709-722.

Eddy, S., Ketkar, A., Zafar, M.K., Maddukuri, L., Choi, J.Y., and Eoff, R.L. (2014). Human Rev1 polymerase disrupts G-quadruplex DNA. *Nucleic Acids Res* *42*, 3272-3285.

Greenberg, R.A., Sobhian, B., Pathania, S., Cantor, S.B., Nakatani, Y., and Livingston, D.M. (2006). Multifactorial contributions to an acute DNA damage response by BRCA1/BARD1-containing complexes. *Genes Dev* *20*, 34-46.

Hansel-Hertsch, R., Beraldi, D., Lensing, S.V., Marsico, G., Zyner, K., Parry, A., Di Antonio, M., Pike, J., Kimura, H., Narita, M., *et al.* (2016). G-quadruplex structures mark human regulatory chromatin. *Nat Genet* *48*, 1267-1272.

Hansel-Hertsch, R., Di Antonio, M., and Balasubramanian, S. (2017). DNA G-quadruplexes in the human genome: detection, functions and therapeutic potential. *Nat Rev Mol Cell Biol* *18*, 279-284.

Hazel, P., Huppert, J., Balasubramanian, S., and Neidle, S. (2004). Loop-length-dependent folding of G-quadruplexes. *J Am Chem Soc* *126*, 16405-16415.

Kisielewska, J., Lu, P., and Whitaker, M. (2005). GFP-PCNA as an S-phase marker in embryos during the first and subsequent cell cycles. *Biol Cell* *97*, 221-229.

Maleki, P., Mustafa, G., Gyawali, P., Budhathoki, J.B., Ma, Y., Nagasawa, K., and Balci, H. (2019). Quantifying the impact of small molecule ligands on G-quadruplex stability against Bloom helicase. *Nucleic Acids Res* *47*, 10744-10753.

Marchand, A., Rosu, F., Zenobi, R., and Gabelica, V. (2018). Thermal Denaturation of DNA G-Quadruplexes and Their Complexes with Ligands: Thermodynamic Analysis of the Multiple States Revealed by Mass Spectrometry. *J Am Chem Soc* *140*, 12553-12565.

Mechali, M. (2010). Eukaryotic DNA replication origins: many choices for appropriate answers. *Nat Rev Mol Cell Biol* *11*, 728-738.

Nicovich, P.R., Owen, D.M., and Gaus, K. (2017). Turning single-molecule localization microscopy into a quantitative bioanalytical tool. *Nat Protoc* *12*, 453-460.

Piazza, A., Adrian, M., Samazan, F., Heddi, B., Hamon, F., Serero, A., Lopes, J., Teulade-Fichou, M.P., Phan, A.T., and Nicolas, A. (2015). Short loop length and high thermal stability determine genomic instability induced by G-quadruplex-forming minisatellites. *EMBO J* *34*, 1718-1734.

Puig Lombardi, E., Holmes, A., Verga, D., Teulade-Fichou, M.P., Nicolas, A., and Londono-Vallejo, A. (2019). Thermodynamically stable and genetically unstable G-quadruplexes are depleted in genomes across species. *Nucleic Acids Res* *47*, 6098-6113.

Rachwal, P.A., Brown, T., and Fox, K.R. (2007). Sequence effects of single base loops in intramolecular quadruplex DNA. *FEBS Lett* *581*, 1657-1660.

Raghunandan, M., Chaudhury, I., Kelich, S.L., Hanenberg, H., and Sobeck, A. (2015). FANCD2, FANCI and BRCA2 cooperate to promote replication fork recovery independently of the Fanconi Anemia core complex. *Cell Cycle* *14*, 342-353.

Ray, S., Qureshi, M.H., Malcolm, D.W., Budhathoki, J.B., Celik, U., and Balci, H. (2013). RPA-mediated unfolding of systematically varying G-quadruplex structures. *Biophys J* *104*, 2235-2245.

Rodriguez, R., Muller, S., Yeoman, J.A., Trentesaux, C., Riou, J.F., and Balasubramanian, S. (2008). A novel small molecule that alters shelterin integrity and triggers a DNA-damage response at telomeres. *J Am Chem Soc* *130*, 15758-15759.

Sarkies, P., Murat, P., Phillips, L.G., Patel, K.J., Balasubramanian, S., and Sale, J.E. (2012). FANCI coordinates two pathways that maintain epigenetic stability at G-quadruplex DNA. *Nucleic Acids Res* *40*, 1485-1498.

Schonenberger, F., Deutzmann, A., Ferrando-May, E., and Merhof, D. (2015). Discrimination of cell cycle phases in PCNA-immunolabeled cells. *BMC Bioinformatics* *16*, 180.

Whelan, D.R., Lee, W.T.C., Marks, F., Kong, Y.T., Yin, Y., and Rothenberg, E. (2020). Super-resolution visualization of distinct stalled and broken replication fork structures. *PLOS Genetics*.

Whelan, D.R., Lee, W.T.C., Yin, Y., Ofri, D.M., Bermudez-Hernandez, K., Keegan, S., Fenyo, D., and Rothenberg, E. (2018). Spatiotemporal dynamics of homologous recombination repair at single collapsed replication forks. *Nat Commun* *9*, 3882.

Wu, C.G., and Spies, M. (2016). G-quadruplex recognition and remodeling by the FANCI helicase. *Nucleic Acids Res* *44*, 8742-8753.

Yin, Y., Lee, W.T.C., and Rothenberg, E. (2019). Ultrafast data mining of molecular assemblies in multiplexed high-density super-resolution images. *Nat Commun* *10*, 119.

Yin, Y., and Rothenberg, E. (2016). Probing the Spatial Organization of Molecular Complexes Using Triple-Pair-Correlation. *Sci Rep* *6*, 30819.

REVIEWER COMMENTS

Reviewer #2 (Remarks to the Author):

I appreciate the efforts that the authors took to respond to my comments and some of my concerns have been addressed but not the three major issues I had with the manuscript.

1. I raised my concern that the effect of FANCI deletion/depletion is seen only when PDS is used, and is not seen under native conditions. If FANCI is required to resolve G4 structures during normal cellular cycles, such effects should be visible even in the absence of PDS. All the authors have done, essentially, in response to this concern is to add 'stable' in front of G4 structures. Without such piece of evidence all they can claim is that FANCI is necessary to resolve PDS-stabilized G4 DNA, and lack of precision in their claim is problematic. Excerpts from my original comments are below.

"It is also very important to show that the fraction of G4 replisome among all replisomes observed increases with FANCI deletion, which is currently lacking in the paper. They only show FANCI's roles when G4 is chemically stabilized but not under native conditions."

" In fact, lack of data showing that FANCI resolves G4 DNA under native condition is concerning. If FANCI is required for G4 resolution, when it is deleted, a higher fraction of replisomes should show G4 signal. Why do they not report these quantities when the measurements are performed without PDS?"

2. The authors claim that G4 DNA is formed between MCM helicase and nascent DNA. I raised concerns about this point (still claimed in the abstract) and requested the following analysis.

"They would need to plot the same but with EdU and G4 to align along the horizontal axis and show that MCM does not appear between the EdU and G4. They would also need to do the same but with MCM and G4 aligned along the horizontal axis and show that Edu does not appear between them."

Such analysis does not require new experiments but the authors appear to have missed my request.

3. I also raised a concern about statistical significance that may potentially arise from the small fraction of G4 containing replication forks. The key question I asked is how many G4 containing forks are observed per single cell image and how does this compare to what's expected.

"However, such a small fraction raises some concerns about the robustness of this important conclusion. How many RFs are observed typically in a single cell when they perform? How many RF-associated G4s are observed per cell. Just from looking at the images, I would imagine this would be a very small number, in single digits if not lower. "

"it will be very helpful to provide a back of the envelope estimation of how many RFs are expected to be observed in a single imaging plane and what fraction of them are expected to contain G4 containing sequences"

Reviewer #3 (Remarks to the Author):

The authors have addressed my concerns. I do not have any remaining comments.

Point-by-Point Response to the Reviewers' Comments for NCOMMS-20-28452A.

Single-Molecule Imaging Reveals Replication Fork Coupled Formation of G-quadruplex Structures Hinders Local Replication Stress Signaling

Wei Ting C. Lee, Yandong Yin, Michael J. Morten, Peter Tonzi, Diana C. Odermatt, Pam Pam Gwo, Mauro Modesti, Sharon B. Cantor, Kerstin Gari, Tony T. Huang, Eli Rothenberg*.

For clarity, the reviewers' comments are in **black** and our responses are marked in **blue**.

Reviewer #2 (Remarks to the Author):

I appreciate the efforts that the authors took to respond to my comments and some of my concerns have been addressed but not the three major issues I had with the manuscript.

1. I raised my concern that the effect of FANCI deletion/depletion is seen only when PDS is used, and is not seen under native conditions. If FANCI is required to resolve G4 structures during normal cellular cycles, such effects should be visible even in the absence of PDS. All the authors have done, essentially, in response to this concern is to add 'stable' in front of G4 structures. Without such piece of evidence all they can claim is that FANCI is necessary to resolve PDS-stabilized G4 DNA, and lack of precision in their claim is problematic. Excerpts from my original comments are below.

"It is also very important to show that the fraction of G4 replisome among all replisomes observed increases with FANCI deletion, which is currently lacking in the paper. They only show FANCI's roles when G4 is chemically stabilized but not under native conditions."

"In fact, lack of data showing that FANCI resolves G4 DNA under native condition is concerning. If FANCI is required for G4 resolution, when it is deleted, a higher fraction of replisomes should show G4 signal. Why do they not report these quantities when the measurements are performed without PDS?"

We thank the reviewer's for reiterating their concerns, however, we emphasize that we have thoroughly addressed these queries in our original response and our revisions therein. Specifically, our original response and revised manuscript include detailed analyses and additional data and measurements relevant to the consequences of FANCI deletion, and FANCI's role in resolving G4s under native (non-PDS-treated) conditions. Below we include our response from the first revision and provide the corresponding texts/ figures reported in our manuscript in orange (with arrows pointing the corresponding data).

- We emphasize that the depletion of FANCI in U2OS cells did result in an enrichment in nuclear G4s, which is shown in Figure S2b. We note that this enrichment was further elevated upon PDS exposure, as shown in Figure S2b (Line 214).

(Supplementary Figure S2b)

- Line 214: “Depletion of FANCJ in U2OS cells resulted in an enrichment in nuclear G4s (Supplementary Fig. S2b), which was further elevated upon PDS exposure. In contrast, the brief PDS treatment did not yield a noticeable change in nuclear G4 signal in siCTRL cells (Supplementary Fig. S2b).”
- To determine whether the observed increase in G4 structures stems from the stabilization of RF-coupled G4s, we used both DBSCAN/NND and TC as independent approaches for quantifying the relative G4-Replisome frequencies in siCTRL and siFANCJ cells. The fractions of G4-positive-replisome in siCTRL vs siFANCJ cells, calculated by DBSCAN/NND, is shown in Figure S2c. We noted that although the nuclear density of G4 shown an increase in NT siFANCJ cells compared to NT siCTRL cells (Figure S2b), the fraction of PCNA associated with G4 is lower in NT siFANCJ cells compared to NT siCTRL cells (Figure S2c).

(Supplementary Figure S2c)

- We therefore quantified the level of active replication via AC analysis of EdU (Figure S2e) and of chromatin bound PCNA (Figure S2f), each serve as a distinct marker of active replications forks. These results revealed an increase in the level of active replication in siFANCJ cells as compared to siCTRL cells. This increase is anticipated due to the previously reported role of FANCJ in regulating origin firing and replication initiation (Greenberg et al., 2006; Raghunandan et al., 2015), where FANCJ-depletion lead to increase in origin firing and active forks, shown as elevated EdU and PCNA levels. The overall increase in active forks would therefore lead to a decrease in the relative G4-replication fraction in siFANCJ cells. The data and above explanations are now included in Figure S2.

(Supplementary Figure S2e and f)

-
- Line 984: “Noted that the fraction of PCNA associated with G4 is lower in NT siFANCJ cells than in siCTRL cells. We reasoned that this is due to the increased amount of overall abundance of active replication forks in when FANCJ is depleted¹. The level of active replication was quantified via SMLM-AC analysis of EdU (e) and of chromatin bound PCNA (f), with each serving as distinct marker of active replications forks. This revealed a ~55-60% increase in the active replication level in NT-siFANCJ cells, as compared to siCTRL cells, supporting that the extra origin firing results in the relative decrease in G4-replication fraction in siFANCJ cells.”

2. The authors claim that G4 DNA is formed between MCM helicase and nascent DNA. I raised concerns about this point (still claimed in the abstract) ...

...and requested the following analysis.

"They would need to plot the same but with EdU and G4 to align along the horizontal axis and show that MCM does not appear between the EdU and G4. They would also need to do the same but with MCM and G4 aligned along the horizontal axis and show that Edu does not appear between them."

Such analysis does not require new experiments but the authors appear to have missed my request.

We would like to clarify that this point was also addressed in our original response, where we provided further explanation and additional experimental data and rationale to support the conclusion that G4 forms between MCM helicase and polymerases. Specifically, we emphasize that by imaging G4 with MCM and nascent DNA (EdU), our SMLM-TC approach revealed a significant association between G4 and replisomes. To test whether these replication fork (RF)-coupled G4 happens within replisomes (i.e. in between MCM and EdU), we designed experiments in which we directly induce the uncoupling of MCM from DNA polymerase by briefly treating cell with aphidicolin (APH), which results in more ssDNA within the replisomes. In agreement with our hypothesis, our SMLM-TC analyses were able to detect the increase of G4-replisomes (Figure 1h and i), indicating that DNA G4s can form at forks during replication progression as the ssDNA is exposed between MCM and nascent DNA.

(Figure 1h and i)

We further note that the triplet pattern tends to present as an equilateral triangle as opposed to a linear configuration (examples shown in Figure 1b and e, or Supplementary Note 3 Figure 1c). This is why plotting the same data with different horizontal axis alignment would not provide any additional information. We elaborate on the plotting of TC triplets in Supplementary Note 3 of the current study, as well as in our previous study (Yin et al., Nature Communications, 2019). We emphasize that throughout the study, the magnitudes of any TC triplets derived from individual nuclei are provided via TC quantifications, whereas the TC triplet plots (overlays) serve as a visual representation. It is important to note that variations in distance configurations and pattern alignment do not affect the quantifications of the specific components within each pattern.

3. I also raised a concern about statistical significance that may potentially arise from the small fraction of G4 containing replication forks. The key question I asked is how many G4 containing forks are observed per single cell image and how does this compare to what's expected.

"However, such a small fraction raises some concerns about the robustness of this important conclusion. How many RFs are observed typically in a single cell when they perform? How many RF-associated G4s are observed per cell. Just from looking at the images, I would imagine this would be a very small number, in single digits if not lower. "

"it will be very helpful to provide a back of the envelope estimation of how many RFs are expected to be observed in a single imaging plane and what fraction of them are expected to contain G4 containing sequences"

We have discussed the observed vs expected in our original response in great detail. Briefly, our imaging and quantification methods detected an average density of 1.25E-6/nm² replication sites per single cell image ($N = 118$), which would be ~853 replication sites in an U2OS nucleus with diameter of ~10 μm . By measuring the fraction of replication sites that are non-randomly colocalized with G4s via the DBSCAN/NND approaches, we estimated an ~2.3% of G4-positive replication sites, which means ~20 G4-containing forks were observed. We emphasize that when performing the DBSCAN/NND analysis, we compared the level of G4-replication colocalization with a "random" distribution, which is generated via randomly repositioning and orientating the clusters generated from DBSCAN segmentation. Despite the small fraction of G4-containing replication sites, our analysis provides that their colocalization is significantly greater than total random colocalization (two-sample t-test). This observed value of 2.3% G4-positive replication sites is also consistent with the estimates provided in Supplementary Note 2.

Reviewer #3 (Remarks to the Author):

The authors have addressed my concerns. I do not have any remaining comments.

We thank this reviewer for his/her support of our manuscript.